

**A global wetland methane emissions and uncertainty dataset for atmospheric chemical transport models.**

A. Anthony Bloom[1], Kevin Bowman[1], Meemong Lee[1], Alexander J. Turner[2], Ronny Schroeder[3], John
R. Worden[1], Richard Weidner [1], Kyle C. McDonald[1,3], Daniel J. Jacob[2].

[1]Jet Propulsion Laboratory, California Institute of Technology, Pasadena, CA, USA

[2]School of Engineering and Applied Sciences, Harvard University, Cambridge, MA, USA

[3]The City College of New York, NY, USA

*Correspondence to*: A. Anthony Bloom ([abloom@jpl.nasa.gov](mailto:abloom@jpl.nasa.gov))

**Abstract.** Wetland emissions remain one of the principal sources of uncertainty in the global atmospheric methane ($CH_4$) budget, largely due to poorly constrained process controls on $CH_4$ production in waterlogged soils. Process-based estimates of global wetland $CH_4$ emissions and their associated uncertainties can provide crucial prior information for model-based top-down $CH_4$ emission estimates. Here we construct a global wetland $CH_4$ emission model ensemble for use in atmospheric chemical transport models. Our $0.5° \times 0.5°$ resolution model ensemble is based on satellite-derived surface water extent and precipitation re-analyses, nine heterotrophic respiration simulations (eight carbon cycle models and a data-constrained terrestrial carbon cycle analysis) and three temperature parameterizations for the period 2009-2010; an extended ensemble subset – based solely on precipitation and the data-constrained terrestrial carbon cycle analysis – is derived for the period 2001-2015. We incorporate the mean of the full and extended model ensembles into GEOS-Chem and compare model against surface measurements of atmospheric $CH_4$; model performance (site-level and zonal mean anomaly residuals) compares favourably against published wetland $CH_4$ emissions scenarios. We find that uncertainties in carbon decomposition rates and wetland extent together account for more than 80% of the primary uncertainty in the timing, magnitude and seasonal variability of wetland $CH_4$ emissions, although uncertainty in the temperature $CH_4$:C dependence is a significant



contributor to seasonal variations in mid-latitude wetland $CH_4$ emissions. The combination of satellite, carbon cycle models and temperature dependence parameterizations provides a physically informed structural a priori uncertainty critical for top-down estimates of wetland $CH_4$ fluxes: specifically, our ensemble can provide enhanced information on the prior $CH_4$ emissions uncertainty and the error

covariance structure, as well as a means for using posterior flux estimates and their uncertainties to quantitatively constrain global wetland $CH_4$ emission biogeochemical process controls.

## 1 Introduction

Methane ($CH_4$) is a potent greenhouse gas, with a global warming potential of more than 25 times that of $CO_2$ on a 100-year time horizon (Myhre et al., 2013). The global $CH_4$ budget and growth rate remain poorly understood, largely due to poorly resolved evolution of atmospheric $CH_4$ sources and sinks (Nisbet et al., 2014). Wetland $CH_4$ emissions are the largest natural source of atmospheric $CH_4$, amounting to roughly 20 – 40% of global $CH_4$ emissions (Ciais et al., 2013). The large disparities

between a range of top-down and bottom up wetland $CH_4$ estimates (Kirschke et al., 2013; Melton et al., 2013) arise from large uncertainties in the timing, distribution and the underlying processes controlling net wetland $CH_4$ production.

In wetland soils, $CH_4$ is produced by the decomposition of organic matter in anaerobic (oxygen depleted) environments. The dominant processes controlling the seasonal and inter-annual variations

include carbon availability (soil C substrate) and decomposition rate, wetland inundation extent, and temperature (Yvon-Durocher et al., 2014). Other important controls on wetland $CH_4$ emissions include the presence of macrophytes (Laanbroek 2010), organic C decomposition rates (Miyajima et al., 1997) and soil pH (Singh et al., 2000), amongst other factors. The link between terrestrial carbon-water cycling and wetland $CH_4$ emissions is of particular interest from a terrestrial greenhouse gas emissions

standpoint: inter-annual variations in terrestrial carbon cycling (Le Quéré et al., 2013) can affect wetland $CH_4$ emissions on seasonal-to-century timescales (Hodson et al., 2011). The role of carbon cycle dynamics in global wetland $CH_4$ emissions is increasingly recognized: temporal variations in gross primary production influence short-term carbon supply (such as carbon inputs from root exudates



and fine litter), as well as long-lived carbon stores (such as wood litter turnover or soil organic C) in wetland soils (Riley et al., 2011; Bloom et al., 2012; Melton et al., 2013). The combined response of $CO_2$ and $CH_4$ fluxes to climatic variability remains poorly characterized. For example, increasing temperatures in boreal ecosystems could lead to higher carbon uptake, increased respiration and drier

soils (Watts et al., 2014), and it is currently unclear whether these processes amount to an amplifying or dampening effect on boreal $CH_4$ emissions. From a greenhouse gas balance standpoint, quantifying the global-scale process links between terrestrial carbon cycling and wetland $CH_4$ emissions is crucial to characterizing the combined terrestrial biosphere $CO_2$ and $CH_4$ flux response to climatic variability.

Quantification of regional wetland $CH_4$ emissions remains challenging. While wetland $CH_4$

emissions are relatively well constrained on a global scale (Kirschke et al., 2013), regional $CH_4$ fluxes are difficult to detect, due to their comparatively diffuse nature – relative to anthropogenic point sources – and the scarcity of direct measurements of wetland $CH_4$ emissions. From a bottom-up perspective, challenges in wetland $CH_4$ modelling stem from order-of-magnitude uncertainties on wetland $CH_4$ emissions factors and their spatio-temporal dependence on biogeochemical process controls.

Nonetheless, for top-down $CH_4$ emission estimates, prior knowledge of wetland $CH_4$ emissions and their associated uncertainty is critical in the formulation of Bayesian atmospheric $CH_4$ inversions. Atmospheric inversions combine $CH_4$ measurements from surface, aircraft and satellites (Wecht et al., 2014b; Jacob et al., 2016) and the prior probability on the magnitude and uncertainty characteristics of $CH_4$ emissions (Bousquet et al., 2011; Pison et al., 2013; Fraser et al., 2014; Turner et al., 2015): these

$CH_4$ inversions typically formulate wetland $CH_4$ emission uncertainty as a spatio-temporally uncorrelated and normally distributed $CH_4$ prior. However, inter-model similarities reveal significant levels of emergent correlations in the timing, magnitude and spatial variability of wetland $CH_4$ emissions. For example, the Wetland CH4 Inter-comparison of Models Project (WETCHIMP) model ensemble (Melton et al., 2013) reveals varying levels of spatial and temporal agreement between

models; these correlations stem from large-scale patterns in biogeochemical process controls (such as temperature, inundation and carbon cycling). Given the relatively large WETCHIMP $CH_4$ emission uncertainties (model range is typically 150-300% of model mean over major wetland areas, and greater elsewhere), this prior 'biogeochemical covariance' can potentially amount to a critical constraint on



atmospheric CH$_4$ inversions: such a covariance structure can be incorporated in an atmospheric inversion cost-function (Michalak et al., 2005) or as a means for improving attribution of posterior CH$_4$ fluxes to wetland CH$_4$ emissions (Wecht et al., 2014a).

Here we propose a process-informed wetland CH$_4$ emission ensemble based on multiple
terrestrial biosphere models, wetland extent scenarios and CH$_4$:C temperature dependencies. In contrast to a conventional process-based model inter-comparison approach, our ensemble statistics are derived by exhaustively combining a range of temperature, carbon and wetland extent parameterizations. An advantage of our approach is that it provides a prior probability distribution of biogeochemical process control uncertainty: our ensemble can be further constrained – based top-down CH$_4$ emission estimates
– to quantify the combined probability distribution of carbon models, CH$_4$:C temperature dependencies and wetland extent scenarios.

We formulate a full (2009-2010) and extended (2001-2015) estimate of wetland CH$_4$ emission magnitude and its associated biogeochemical covariance structure, based on knowledge of the global wetland CH$_4$ source and the primary biogeochemical process controls. We validate and compare the
wetland CH$_4$ emissions ensemble against a suite of regional studies, and we use a global atmospheric chemical transport model (GEOS-Chem, Bey et al., 2001) to evaluate the CH$_4$ emissions ensemble relative to existing wetland CH$_4$ emission models (sections 2 and 3). Finally, we summarize the strengths and limitations of our wetland emissions ensemble and outline its potential applications in global atmospheric inversion frameworks (section 4).

## 2. Wetland CH$_4$ model ensemble

The wetland CH$_4$ emissions ensemble provides CH$_4$ fluxes and associated uncertainty estimates based on four wetland extent parameterizations, nine terrestrial biosphere models of heterotrophic
respiration and three CH$_4$:C temperature parameterization. Global monthly 0.5°×0.5° emissions and their associated uncertainty structure span 2009-2010 (full ensemble, henceforth FE); we also evaluate a subset of the model ensemble spanning 2001-2015 (extended ensemble, henceforth EE). We validate FE



and EE emissions against a range of regional $CH_4$ emission estimates, and we test the updated GEOS-Chem model against 104 surface $CH_4$ measurement sites.

## 2.1 Wetland $CH_4$ emissions & uncertainty

We derive wetland $CH_4$ emissions $F$ $(mg\ CH_4\ m^{-2}\ day^{-1})$ at time $t$ and location $x$ as:

$$F(t,x) = s\,A(t,x)R(t,x)q_{10}^{\frac{T(t,x)}{10}} \qquad (1)$$

where $A(t,x)$ is the wetland extent fraction, $R(t,x)$ is the total C heterotrophic respiration at time for a unit area at time $t$, $q_{10}^{T(t,x)/10}$ is the temperature dependence of the ratio of C respired as $CH_4$ (where $q_{10}$ represents the relative $CH_4$:C respiration for a 10°C increase) and $s$ is a global scale factor. This empirical parameterization provides first order constraints on the role of carbon, water and temperature variability on the global spatial and temporal variability of wetland $CH_4$ emissions. Variants of the

equation 1 parameterization have been used within a range of wetland $CH_4$ emission models (e.g., Hodson et al., 2011, Pickett-Heaps et al., 2011, Melton et al., 2013 amongst others).

In our approach, wetland $CH_4$ emissions statistics within each 0.5°×0.5° gridcell are derived based on a 108-member ensemble of wetland $CH_4$ emission simulations. The 108-member FE is based on 3 $CH_4$:C temperature dependencies, 9 heterotrophic respiration configurations and 4 wetland extent

scenarios (3×9×4 = 108); the six-member EE ensemble is a subset of FE, based on data availability during 2001-2015 (see table 1 for summary). For each ensemble member, the magnitude and uncertainty of $s$ is optimized to match global estimates of the annual wetland $CH_4$ source (Kirshke et al., 2013).

The heterotrophic respiration configurations are derived from 8 terrestrial biosphere models used

in the Multi-scale Synthesis and Terrestrial Model Intercomparison Project (MsTMIP BG1 simulations, see Huntzinger et al., 2013 and Wei et al., 2014 for model and experiment details) and the global CARbon DAta-MOdel fraMework (CARDAMOM) terrestrial carbon analysis (Bloom et al., 2016). V1.0 outputs from the MsTMIP are available for the period 1900-2010 (Huntzinger et al., 2015); the





CARDAMOM analysis was extended to span 2001-2015 based on the Bloom et al., (2016) methodology (see Appendix A for details). Since MsTMIP and CARDAMOM respiration estimates vary intrinsically as a function of temperature, $q_{10}$ only accounts for the temperature dependence of the fraction of C respired as $CH_4$. We prescribe three $CH_4$:C temperature dependencies (table 1) which are

broadly equivalent to a ±50% range on the $CH_4$:$CO_2$ temperature dependence reported by Yvon-Durocher et al., (2014).

Here we use two spatial (i = 1,2) and two temporal (j = 1,2) wetland extent parameterizations approaches to represent the uncertainty associated with the role of hydrology on wetland $CH_4$ emissions. We parameterize wetland extent as the product of a static extent constraint $w_i(x)$ and a

normalized time-varying scale factor $h_{ij}(x,t)$, in the following manner

$$A(x,t) = w_i(x)h_{ij}(x,t) \, , \tag{2}$$

$w_1(x)$ is the sum of all GLOBCOVER wetland and freshwater land cover types (all flooded,

water-logged, and inland water body land-cover types; Bontemps et al., 2011) and $w_2(x)$ the Global Wetland and Lakes Database (GLWD) maximum recorded wetland and freshwater body extent map by Lehner & Doll (2004).

For $\mathbf{h}_{*j}(x,t)$, we use (a) the Surface WAter Microwave Product Series (SWAMPS) multi-satellite surface water product (Schroeder et al., 2015; j=1), and (b) monthly ERA-interim precipitation

(j=2): for i = 1 (i = 2), $h_{ij}(x,t)$ is normalized such that mean (maximum) $h_{ij}(x,t)$ is equal to 1. We note that the two hydrological proxies provide contrasting advantages and disadvantages. Satellite-retrieved surface water extent provides an observation-based constraint on the spatial and temporal extent of wetlands and freshwater bodies. While our temporal scaling of static wetland and freshwater extent mitigates the role of spatial biases in satellite-retrieved inundation, vegetation cover remains a major

confounding variable in satellite-constrained wetland extent (Schroeder et al., 2015). Moreover, satellites cannot directly observe subsurface soil saturation, even though these soils amount to significant $CH_4$ fluxes to the atmosphere (Turetsky et al., 2014). On the other hand, precipitation does not provide a direct constraint on the wetland and freshwater extent; however, it provides an aggregate





constraint on ecosystem hydrological variability and wall-to-wall coverage across the globe. We henceforth refer $F$ as "wetland $CH_4$ emissions"; however, we recognize that lakes, rivers and reservoirs account for ~20% of the total wetland and freshwater body extend (Lehner and Döll, 2004). We discuss the implications of including non-wetland freshwater bodies in $w_i(x)$ section 4.

For each of the 108 FE ensemble configurations (c = 1 − 108), and 6 EE ensemble configurations (c = 1-6), we derive $s_c$ such that:

$$s_c = \frac{G}{\Sigma_x \Sigma_t F_{x,t,c} A_x \frac{\Delta t}{n}} \qquad (3)$$

where $F_{x,t,c}$ are the $c^{th}$ ensemble member fluxes at grid-cell $x$ and time $t$, $A_x$ is the area of grid-cell $x$, $\Delta t$ is the timestep (1 month), $n$ is the number of years, and $G$ is the global total $CH_4$ emitted from wetlands. We derive $s_c$ such that each ensemble members are scaled to $G$ = 175 Tg $CH_4$ yr$^{-1}$ throughout the duration of FE (2009-2010): this is consistent with the Kirschke et al., (2013) mean 2000-2009 top-down wetland $CH_4$ emission estimates (175 Tg $CH_4$ yr$^{-1}$).

We derive the $CH_4$ flux ensemble mean and 5th – 95th percentile ranges at location $x$ and time t based on all 108 $F_{x,t,c}$ ensemble members. For the percentile range calculations, we propagate the global mean wetland $CH_4$ emission uncertainty by Kirschke et al., (2013) (minimum-to-maximum range = 142 – 208 Tg $CH_4$ yr$^{-1}$, or 175 Tg $CH_4$ yr$^{-1}$ ±19%). We create an expanded ensemble (FE$_{exp}$) by randomly perturbing all 108 model $s_c$ values by a factor of U(0.81,1.19) 1000 times, where U() denotes

a random number sampled from a continuous uniform distribution spanning the bracketed numbers. We use the expanded ensemble FE$_{exp}$ to derive the ensemble's spatiotemporal error covariance structure: the quantitative derivation and qualitative interpretation of the error covariance structure is fully described in appendix B.

       We attribute the uncertainty of the timing and magnitude of $F(x,t)$ (maximum $CH_4$ emission

25 month, mean $CH_4$ emissions and $CH_4$ emission variability) to carbon decomposition, wetland extent and $CH_4$:C temperature dependence uncertainty; the derivation of the "primary uncertainty" within each



zonal band (i.e. the dominance of carbon, water or temperature as the dominant source of uncertainty) is
fully described in Appendix C.

## 2.2 GEOS-Chem atmospheric CH$_4$ simulations

We evaluate the FE and EE wetland CH$_4$ emission means against the World Data Centre for
Greenhouse Gases (WDCGG) CH$_4$ measurement sites by incorporating these into the 4°×5° resolution
GEOS-Chem atmospheric chemical and transport model (version 10.01; *acmg.seas.harvard.edu/geos*).
We benchmark the FE and EE runs against GEOS-Chem simulations with the GEOS-Chem wetland
10  CH$_4$ emission inventory (Pickett-Heaps et al., 2011; henceforth GC) and the Bloom et al., (2012)
satellite-constrained wetland emissions (henceforth BL), as these emission estimates have been in a
range of atmospheric chemical transport model simulations (Fraser et al., 2013; Turner et al., 2015;
Wilson et al., 2016 amongst others). We perform each GEOS-Chem forward run for the period 2009-
2010 with a four-year (2005-2009) spin-up period. The non-wetland CH$_4$ sources in GEOS-Chem
consist of biofuel, fossil fuel, livestock, waste, Rice (EDGAR v4.2; European Comission, 2011), fires
(Global Fire Emissions Database version 4; van der Werf et al., 2010), soil C sinks and termites (Fung
et al., 1991) .The non-wetland CH$_4$ fluxes are the same in each run, with the exception of rice source in
run BL (as global wetland and rice emissions are treated as one source by Bloom et al., 2012). While
model CH$_4$ surface concentrations are strongly influenced by wetland CH$_4$ magnitude, timing and
distribution (Bloom et al., 2012, Meng et al., 2015), comparisons between GEOS-Chem outputs and
surface CH$_4$ measurement may also be affected by errors in non-wetland CH$_4$ emissions and in
transport. However, Wecht et al. (2012) and Turner et al. (2015) show that the GEOS-Chem emissions
and transport provide an unbiased representation of the observed latitudinal background. The global
inversion of Turner et al. (2015) using GEOS-Chem emissions as prior further shows no large errors in
non-wetland emissions that would confound the analysis presented here.

For each of the four runs (FE, EE, GC and BL), we use the Wecht et al., (2014a) 1[st] Jan 2005
initial conditions for atmospheric CH$_4$ concentrations in GEOS-Chem. For each simulation, we
performed a four-year spin-up period (2005-2009) using 2009 emissions to reduce the potential





inconsistency between initial conditions and the global distribution of wetland $CH_4$ emissions; this spin-up ensures that the relative variations in Jan 1st 2009 $CH_4$ concentrations for each run are broadly consistent with each emission scenario. We save GEOS-Chem atmospheric $CH_4$ concentrations every 3 hours. We compare mean monthly GEOS-Chem output against all WDCGG sites below 500m altitude

(104 sites with monthly 2009-2010 data in total); this minimizes the topographic mismatch due to vertical $CH_4$ representation in the model. For each site, the nearest 4°×5° GEOS-Chem grid-cell is used for comparison.

### 3. Results, Comparison and Validation

Mean full ensemble (FE) global wetland emissions are largely accounted for by three high-latitude regions, three tropical regions, and sub-tropical southeast Asia (Figure 1a). North America, Scandinavia and Siberia median (5th – 95th percentiles) $CH_4$ fluxes amount to 8% (3 – 30%), 2% (0 – 6%) and 2% (1 – 6%) of global emissions; Amazon wetland emissions amount (30%; 21 – 38%) account for the

largest tropical emission source, followed by the Indonesian archipelago (13%; 7 – 22%), and central Africa (12%; 7 –24%); subtropical southeast Asia emissions account for 5% (1 – 10%). High-latitude (>50°N) and tropical emissions amount to 11% (4 – 29%) and 68% (43 – 84%) of global wetland $CH_4$ emissions, respectively. High-latitude FE emissions exhibit a peak at 60°N (Figure 1b) similar to the GEOS-Chem wetland $CH_4$ emissions inventory (Pickett-Heaps et al., 2011; GC) and further north than

the Bloom et al., (2012) emissions (BL). Tropical emissions for all three emission datasets peak within 0° – 5°S. A comparison between zonal mean emissions (Figure 2) reveals differences of less than 1 Tg/yr/°lat between FE and the extended ensemble (EE) s; the FE zonal mean is comparable to the BL in the near-equatorial tropics and significantly (with respect to the FE model ensemble 90% confidence range) lower everywhere else; the FE zonal mean is comparable to GC in high-latitude and temperate

regions, but significantly lower than GC in the tropics and southern hemisphere.

All $CH_4$ emission models show similar patterns in the temporal distribution of $CH_4$ emissions in high-latitude and temperate regions (with $CH_4$ emissions peaking between July and September, Figure 3).



We note that the larger $CH_4$ fluxes in the BL emissions over Asia and Oceania are due to rice paddy $CH_4$ emissions. The $CH_4$ emission models exhibit 1-month differences in the timing of maximum seasonal $CH_4$ emission across the high-latitudes (generally between June and August). In tropical South America 0° – 20°S latitudes, FE and EE emissions peak March –May, which is comparable to BL

(March); and overall earlier than GC (5°S – 20°S emission peak in September). There is a considerable disagreement between northern tropical Africa emission variability amongst all models. Subtropical Asia FE and EE emissions (20°N – 30°N) peak in June-August, earlier than BL emissions (August-September) and comparable to GC emissions (June).

We compare mean FE and EE (2009-2010) wetland emissions against a range of independent wetland $CH_4$ regional emission estimates (Figure 4). Emissions from Siberian wetlands (Glagolev et al., 2011) Hudson bay lowlands (Pickett-Heaps et al., 2011), and Amazon river basin (Melack et al., 2004) are within 25th – 75th percentile estimates of FE and EE wetland $CH_4$ emissions; Chang et al., (2014)(May-September 2012) wetlands are higher (2.1 Tg $CH_4$ yr$^{-1}$) but within the 5th -95th percentile range of FE

and EE wetland $CH_4$ emission estimates. BL (2009-2010) and GC (2009-2010) estimates are also within 5th – 95th percentile ranges. With the exception of Amazon river basin estimates, the FE and EE emission estimate uncertainty is larger than the Melton et al., (2013) wetland $CH_4$ emission model (WETCHIMP 1993-2004) range. We note the temporal mismatch between the wetland emission estimates shown in Figure 4: however, we expect inter-annual variation in wetland $CH_4$ emissions (e.g.

Bloom et al., 2010; Melton et al., 2013) to be substantially smaller than the FE and EE estimate uncertainty.

On a zonal basis, the "primary uncertainty" – i.e. the dominant source of uncertainty within each band – in mean $CH_4$ emissions and the timing of maximum $CH_4$ emissions is almost completely dominated

carbon decomposition and wetland extent uncertainties (Figure 5). Seasonal variability of $CH_4$ emissions is also largely dominated by carbon and extent uncertainties, although the temperature $CH_4$:C dependence is the dominant source of uncertainty in temperate latitudes. At latitudes > 20°N, wetland extent is the dominant source of uncertainty in mean $CH_4$ emissions, while temperature $CH_4$:C



dependence accounts for less than 10% of the primary uncertainty attribution. At all latitudes > 10°S, carbon decomposition accounts for the vast majority of the primary uncertainty in the timing of wetland $CH_4$ emissions.

We summarizing the $FE_{exp}$ global error covariance structure as an error correlation matrix between mean monthly 2009-2010 emissions across boreal & arctic (>55°N) temperate (23°N – 55°N), tropical (23°S – 23°N) and southern hemisphere (<23°S) latitudes (Figure 6); the error correlation matrix quantitatively summarizes similarities in the spatial and temporal patterns between ensemble members, relative to the ensemble mean (see appendix B for description and interpretation). The $FE_{exp}$ error
correlation matrix highlights positively correlated ensemble member $CH_4$ emissions within each region, with larger correlations (generally r>0.8) between emissions separated by 1-2 months; tropical emissions exhibiting the largest overall temporal correlations (r>0.5). Tropical and boreal & arctic $CH_4$ emissions are overall anti-correlated, however no temporal patterns emerge between these anti-correlations.

Mean 2009-2010 observed and GEOS-Chem forward model run $CH_4$ concentrations (with FE, EE, BL and GC wetland emissions) are broadly consistent on a latitudinal basis. The observed and modelled zonal atmospheric $CH_4$ concentration anomaly (relative to mean global 2009-2010 $CH_4$ concentrations) is shown in Figure 7 (zonal profile root-mean-square errors – RMSE – are 6.4 ppb, 6.6 pbb, 8.4 ppb, 9.2
ppb for FE, EE, BL and GC relative to the observed $CH_4$ anomaly zonal profile). Within the primary wetland $CH_4$ emission latitudes (10°S – 80°N; Figure 2), all mean $CH_4$ model estimates are within the mean standard deviation of observed $CH_4$, except GC at >60°N and all models at 80°N. The median site-level correlation (Pearson's r) between observed and model de-trended $CH_4$ concentrations for FE (0.79; 5th – 95th percentiles = -0.15 - 0.97) is highest, followed by EE (0.77; 5th - 95th percentiles = -
0.24 - 0.96), BL (0.75; 5th - 95th percentiles = -0.08 - 0.99) and GC (0.72; 5th - 95th percentiles = -0.24 - 0.93). The median RMSE between observed and model de-trended $CH_4$ concentrations for FE (10.98 ppb; 5th - 95th percentiles = 2.81 - 53.11 ppb) and EE (11.61ppb 5th - 95th percentiles = 4.01 - 51.69) are lower than BL (12.29 ppb; 5th - 95th percentiles = 2.35 - 49.31 ppb) and GC (median =



12.95 ppb; 5th - 95th percentiles = 5.06 - 50.98). FE and EE improvements (relative to GC and BL Pearson's r and RMSE) are primarily in northern hemisphere high-latitudes latitudes (>50°N; Figure 8). FE Pearson's r and RMSE suggest better performance than GC in southern hemisphere extra-tropical latitudes (<23°S); BL is comparable to FE and outperforms EE across southern hemisphere latitudes.

## 4. Discussion

### 4.1 Model limitations

Densely vegetated wetland areas are likely to amount to a large component of the global wetland $CH_4$ sources; high-carbon density (and high temperatures in the case of tropical wetlands) result in high $CH_4$ emissions under inundated conditions. However, satellite-derived observations of surface water area (Schroeder et al., 2015) are ill-equipped to observe densely vegetated wetland areas, as the passive microwave sensors become increasingly sensitive to vegetation moisture within high biomass
ecosystems (Sippel et al., 1994). For example, estimates of Amazon river basin wetland $CH_4$ emissions range between 16% - 29% (5th – 95th percentiles) of the global wetland emissions source; high biomass density in this region (Saatchi et al., 2011) may be a significant source of inundation area bias. Therefore, while we incorporate prior information on the mean and maximum wetland extent to scale the satellite-derived inundation fraction, we anticipate that errors in seasonal and inter-annual
inundation variability are likely to be larger within densely vegetated wetland areas. We are optimistic that current and upcoming missions such as SMAP and BIOMASS (Entekhabi et al., 2010; Le Toan et al., 2011) combined with data integration approaches (Schroeder et al., 2015; Fluet-Chouinard et al., 2015) can potentially provide additional constraints required to extend current inundation datasets and to improve current surface inundation detection capabilities.

The MsTMIP model ensemble provides a first-order estimate of the magnitude and variability of C decomposition within each 0.5°×0.5° grid-cell. Here we highlight 4 potentially major sources of error: (a) differences in aerobic:anaerobic turnover rates of major (labile and recalcitrant) C pools (b)



systematic differences in wetland and non-inundated area carbon uptake within each 0.5°×0.5° grid-cell, (c) systematic differences in dead organic matter C stocks and accumulation between wetland and non-inundated areas, and (d) lateral flows of C into (or out of) wetland areas. Top-down estimates of seasonal and inter-annual terrestrial $CO_2$ fluxes (e.g. Liu et al., 2014) could be used to independently

assess the validity of heterotrophic respiration from the MsTMIP models and CARDAMOM. In turn, top-down $CH_4$ and $CO_2$ flux retrievals, and range of in-situ and regional-scale $CH_4$ flux estimates (Schriel-Uijl et al., 2011; Chang et al., 2014; Budishchev et al., 2014; amongst others) can be combined to assess whether our empirical parameterization is able capture regional, seasonal and inter-annual wetland $CH_4$ emission variability and their link to the broader terrestrial carbon cycle. Finally, in

succession to eddy covariance tower site analyses of $CO_2$ respiration dependence on temperature (Mahecha et al., 2010), we anticipate that $CH_4$ eddy covariance measurements will provide critical site-level constraints on the temperature dependence of wetland $CH_4$ emissions.

Rice paddies likely amount to <20% of wetland $CH_4$ emissions, and the majority of rice paddy areas are

implicitly excluded from our analysis: GLOBCOVER distinguishes between natural and irrigated water bodies, and GLWD explicitly excludes rice paddy extents in China (which alone accounts for a large portion of global rice paddy $CH_4$ emissions). However, satellite-based inundation fraction retrievals are unable to distinguish the temporal variability of co-located agriculture and natural wetland inundation extent; moreover 0.5°×0.5° carbon cycle model resolution may be insufficient to resolve spatial

differences in wetland and agricultural C cycling. Inadvertent inclusion of co-located rice $CH_4$ emissions is therefore a potential source of bias in our approach. We note that the distinction between wetland and rice $CH_4$ emissions has yet to be consistently addressed in global wetland $CH_4$ emission quantification efforts (see Bloom et al., 2010; Hodson et al., 2011; Melton et al., 2013, and references therein).

$CH_4$ production in non-wetland freshwater bodies, such as very small ponds (Holgerson and Raymond 2016), lakes (Wik et al., 2016) and rivers (Bastviken et al., 2011) is potentially a significant – albeit highly uncertain – term in the global $CH_4$ budget (Kirshke et al., 2013; Bridgham et al., 2013). Our





approach implicitly accounts for non-wetland freshwater body emissions, since their extent is incorporated in grid-cell scaling factors (see Eq. 2). We recognise the challenge in explicitly distinguishing between wetlands and non-wetland freshwater body $CH_4$ emissions, as well as the associated physical and biogeochemical process controls: the spatial and temporal characterization of

wetland and non-wetland freshwater extent remains challenging from the current spatial resolution (~25km) of surface inundation retrievals (Prigent et al., 2007; Schroeder et al., 2015). Equally, from a carbon perspective, $0.5° \times 0.5°$ carbon cycle model resolution is insufficient to resolve spatial variability in wetland and non-wetland freshwater body extent (Lehner and Döll, 2004). Contingent on future resolution enhancements in surface inundation and carbon cycle models, we recommend further

investigation on the adequate distinction and estimation of non-wetland freshwater $CH_4$ emissions for atmospheric $CH_4$ chemical transport modelling applications.

By constraining global emission estimates to the Kirshke et al., (2013) model range, our approach does not challenge the global annual $CH_4$ source and uncertainty (175 Tg $CH_4$ $yr^{-1}$; range = 142 – 206 Tg

$CH_4$ $yr^{-1}$ or ±19%); rather, it places constraints on spatial and temporal wetland $CH_4$ source variability. Since the global uncertainty is substantially smaller than regional, zonal and grid scale uncertainties (Figures 1,2 and 4), we highlight that new or improved constraints on the global wetland $CH_4$ source are unlikely to substantially influence our regional or grid-scale $CH_4$ flux confidence range estimates.

*4.2 Applications*
Based on comparisons against measured $CH_4$ concentrations and a range of regional and global $CH_4$ emission estimates (Figures 2-4, 7-8), we have shown that the FE and EE wetland $CH_4$ emission ensembles robustly represent the global magnitude and uncertainty of wetland $CH_4$ emissions. The ensemble configurations of inundation extent, carbon decomposition and temperature dependence have

together provided a characterization of the dominant source of uncertainty in global wetland $CH_4$ estimates (Figure 5). The approach outlined here provides a framework for producing prior emission estimates and associated uncertainty. The error covariance structure – along with the $CH_4$ observing system capabilities (Wecht et al., 2014a) – can be used to devise an optimal strategy for spatially and/or





temporally aggregating $CH_4$ fluxes in an atmospheric inversion framework. Retrieved $CH_4$ flux from assimilating atmospheric $CH_4$ observations in an inverse modelling framework (e.g. Fraser et al., 2013) could in turn provide a quantitative constraint on the wetland ensemble: the FE and EE model members can be treated as an ensemble of probable biogeochemical process hypotheses that can be weighted

against atmospheric constraints. In contrast to conventional wetland $CH_4$ emission estimates (Riley et al., 2011; Pickett-Heaps et al., 2011) and model inter-comparisons (Melton et al., 2013), top-down $CH_4$ flux estimates can constrain the joint probability distribution of FE and EE carbon models, wetland extent parameterizations, and temperature dependencies.

We anticipate extensions of the FE beyond the 2009-2010 time period, contingent on the extensions of the MsTMIP and SWAMPS dataset beyond 2010 and 2012 respectively. In light of continued satellite $CH_4$ retrievals from GOSAT (Parker et al., 2011; Butz et al., 2011) and upcoming satellite $CH_4$ measurement from TROPOMI on-board ESA Sentinel 5 precursor (Veefkind et al., 2012), we anticipate that the FE and EE datasets will provide key process-based prior knowledge in future atmospheric $CH_4$

inversions.

### Appendix A: CARDAMOM extension

CARDAMOM heterotrophic respiration was derived from the Bloom et al., (2016) global terrestrial C cycle 1°×1° analysis. CARDAMOM retrieved C state and process variables for the period 2001-2010 were used to run the ecosystem carbon balance model DALEC2 (Bloom & Williams 2015) to span 2001-2015. The 2011-2015 ERA-interim meteorological drivers and MODIS burned area were obtained as described by Bloom et al., (2016). The CARDAMOM output consists of 4000 heterotrophic

respiration realisations at each monthly time-step: for each time-step, we use the median CARDAMOM heterotrophic respiration output. We downscale the data to a 0.5°×0.5° resolution using a nearest neighbour interpolation.





**Appendix B: Error correlation structure**

We derive the model ensembles' space-time $n \times n$ error correlation matrix **M** as follows:

$M_{ij} = \mathrm{cor}(\mathbf{A}_{i,*} \mid \mathbf{A}_{j,*})$                                    (A1)

where $n$ corresponds to the number of space and time wetland $CH_4$ emission aggregations, and i, j span $1-n$,. $A_{(i,m)}$ and $A_{(j,m)}$ correspond to the total $CH_4$ flux for model $m$ within the $i$th and $j$th space-time aggregations (i.e. total wetland $CH_4$ emissions within a given time & area); $\mathbf{A}_{i,*}$ and $\mathbf{A}_{j,*}$ are $1 \times N$

vectors, where $N$ is the number of models within the ensemble.  the "cor()" operator denotes the Pearson's correlation coefficient between the two bracketed vectors. For Figure 6, $A_{1,m}$ we aggregate model wetland $CH_4$ emissions for each month across four zonal bands: Boreal & Arctic (>55°N) Temperate (23°N – 55°N), Tropical (23°S – 23°N) and Southern Hemisphere (<23°S). **Interpretation:** a perfect correlation between the $i$th and $j$th indices ($M_{ij}$ =1) indicates that models are consistently over-

or under-predicting $CH_4$ emissions at times-and-locations $i$ and $j$ relative to the ensemble mean; a perfect anti-correlation ($M_{ij}$ =-1) indicates that models consistently over-predicting $CH_4$ emissions at time-and-location $i$ consistently under-predict $CH_4$ emissions at time-and-location $j$ (relative to the ensemble mean) and vice versa.

**Appendix C: Primary process uncertainty**

We quantify the primary process uncertainty of wetland $CH_4$ emission state variables ($s$ = 1-3; 1. maximum emission month, 2. mean $CH_4$ emissions and 3. seasonal variability (standard deviation)) to wetland emission controls  ($e$ = 1-3; 1. model carbon decomposition, 2. $CH_4$:C temperature dependence

and 3. wetland extent parameterization) at location $x$ as follows:

$$R_{x,s,e} = \sum_{c=1}^{N} \frac{\max(\mathbf{M}_{x,s,m_c}) - \min(\mathbf{M}_{x,s,m_c})}{N}$$                                    (A2)





where $R_{x,s,e}$ is the mean range of state variable $s$ across the ensemble given a fixed emission control $e$; $\mathbf{M}_{x,s,*}$ is a vector of all ensemble member state variables $s$ at location x; $\mathbf{m}_c$ denotes the indices of ensemble subset driven by $c$th emission control $e$; $N$ are the number of configurations for each $e$ (the ensemble configuration details are show in Table 1). For example, $R_{100,1,1}$ is the mean range of seasonal

CH4 variability ($s=3$) for a fixed carbon model configuration ($e=1$) at the $100^{\text{th}}$ gridcell ($x=100$). We attribute the zonal primary uncertainty of state variable $s$ to emission control $e$ as:

$$P_{z,s,e} = \frac{\sum_{x_z} r_{x_z,s,e}\, F_{x_z}}{\sum_{x_z} F_{x_z}} \times 100\% \qquad\qquad (A3)$$

where $x_z$ are the pixels $x$ within 5° zonal band $z$, $F_{x_z}$ is the mean 2009-2010 area-integrated CH4 flux (Eq. 1 in main text). $r_{x_z,s,e} = 1$ if $R_{x_z,s,e} = \min(\mathbf{R}_{x_z,s,*})$ otherwise $r_{x_z,s,e} = 0$; the "min()" function denotes the minimum element of the bracketed vector; i.e. $e$ is the largest source of uncertainty when the mean range in state variable $s$ is smallest for a fixed $e$. $P_{z,s,e}$ denotes the percentage of zonal band $z$ where emission control $e$ is the greatest source of uncertainty for each $s$.

### Data availability

The full ensemble (FE), extended ensemble (EE) presented here are currently available upon request and will become publicly available through the NASA DAAC. MsTMIP monthly 0.5°×0.5° datasets

were obtained from *nacp.ornl.gov/MsTMIP.shtml*. ERA-interim datasets were obtained from *apps.ecmwf.int/datasets/data/interim-full-mnth*. CARDAMOM 2001-2010 heterotrophic respiration outputs are available at *datashare.is.ed.ac.uk/handle/10283/875*; 2011-2015 heterotrophic extensions outputs are available upon request. Inundation datasets were obtained from *wetlands.jpl.nasa.gov*. The GLWD dataset was obtained from *gcmd.gsfc.nasa.gov*. The GLOBCOVER dataset was obtained from

*due.esrin.esa.int*. The WDCGG data was obtained from *ds.data.jma.go.jp/gmd/wdcgg*.

### Acknowledgments



*Part of this research was carried out at the Jet Propulsion Laboratory, California Institute of Technology, under a contract with the National Aeronautics and Space Administration. Funding for this study was provided through a NASA Carbon Monitoring System Grant #NNH14ZDA001N-CMS.*

*Funding for the Multi-scale synthesis and Terrestrial Model Intercomparison Project (MsTMIP; http://nacp.ornl.gov/MsTMIP.shtml) activity was provided through NASA ROSES Grant #NNX10AG01A. Data management support for preparing, documenting, and distributing model driver and output data was performed by the Modeling and Synthesis Thematic Data Center at Oak Ridge National Laboratory (ORNL; http://nacp.ornl.gov), with funding through NASA ROSES Grant*

*#NNH10AN681. Finalized MsTMIP data products are archived at the ORNL DAAC (http://daac.ornl.gov).*

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

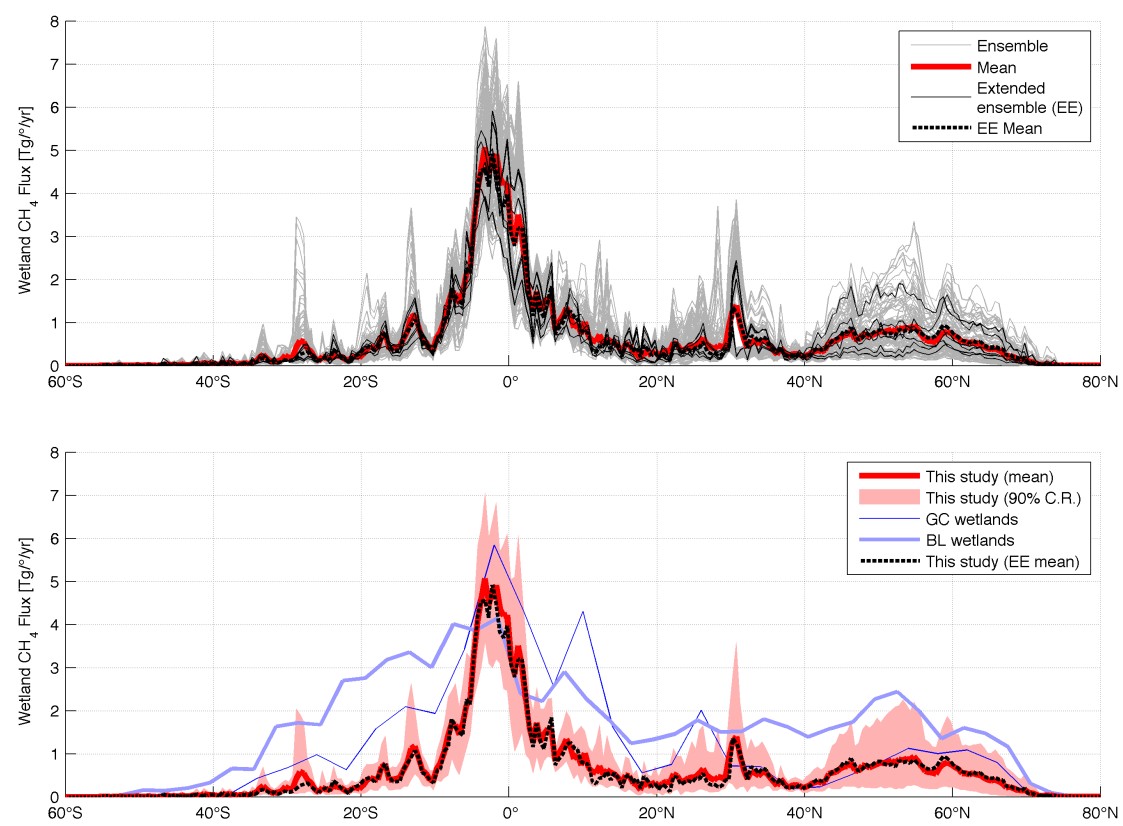

**Figure 2. Top:** Zonal profile of full ensemble (FE, red) extended ensemble (EE, black dashed line) and mean wetland CH$_4$ emissions from 108 ensemble members (grey). **Bottom:** mean FE (red), mean EE (black dashed line) and 90% FE confidence range (pink), GEOS-Chem wetland emission inventory (GC) and the Bloom et al., (2012) emissions (BL).



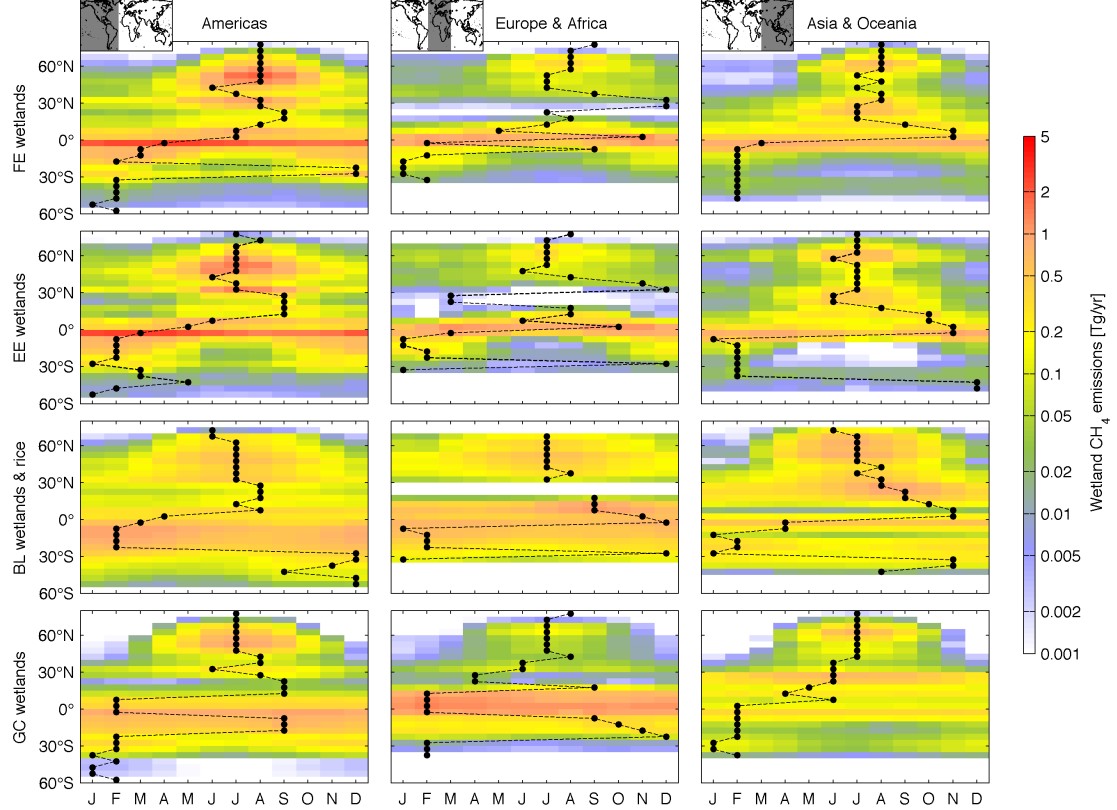

**Figure 3.** Seasonal mean zonal profiles for this study (full ensemble: FE; extended ensemble: EE) Bloom et al., (2012) wetland emissions (BL) and GEOS-Chem wetland emissions inventory (GC) for North & South America (left column; 180°W – 35°W); Europe & Africa (center column; 35°W – 55°E) and Asia & Oceania (right column; 55°E – 180°E). The black dotted line denotes the maximum emissions month within each 5° latitude bin.



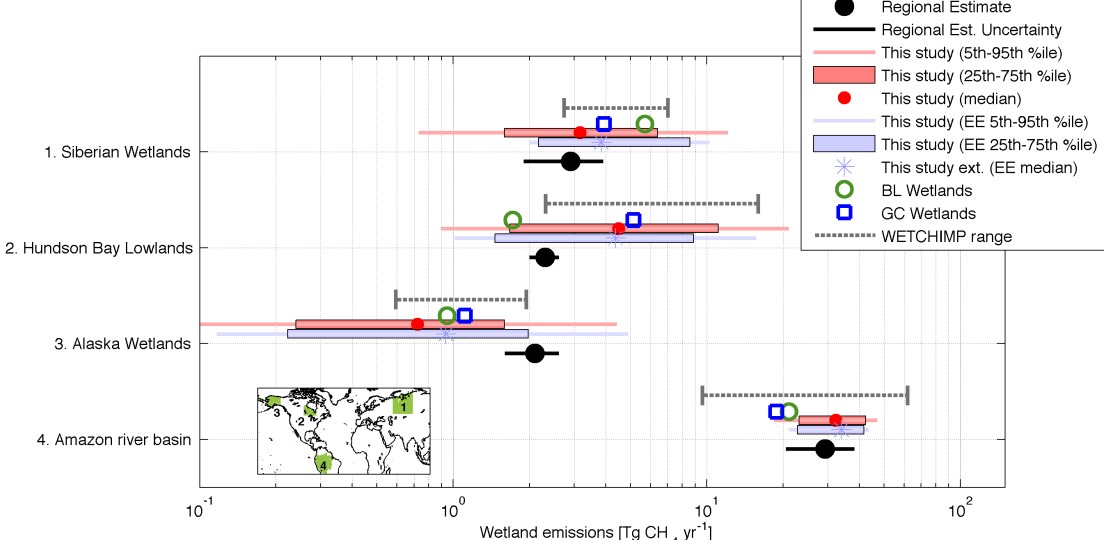

**Figure 4.** Comparison between mean annual regional wetland CH$_4$ emission estimates (1. Glagolev et al., 2011; 2. Picket Heaps et al., 2011; 3. Chang et al., 2014; 4. Melack et al., 2004) and global wetland emission datasets by Bloom et al., 2012 emissions (BL), the GEOS-Chem wetland CH$_4$ emission inventory (GC); this study (full ensemble: FE; extended ensemble: EE), and the range of WETCHIMP models (Melton et al., 2013). The regions (1-4) are shown in the inset map.



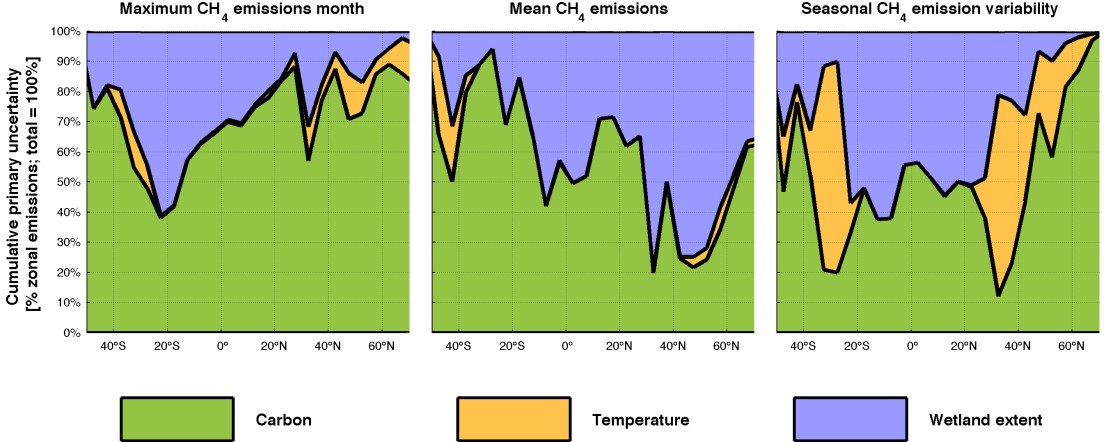

**Figure 5.** Primary uncertainty attribution of maximum $CH_4$ emissions month (left), magnitude (center) and seasonal variability (right), to carbon decomposition, temperature $CH_4$:C dependence ($q_{10}$) and wetland extent parameterization, within 5° latitude bins. The derivation of primary uncertainties is described in Appendix C.





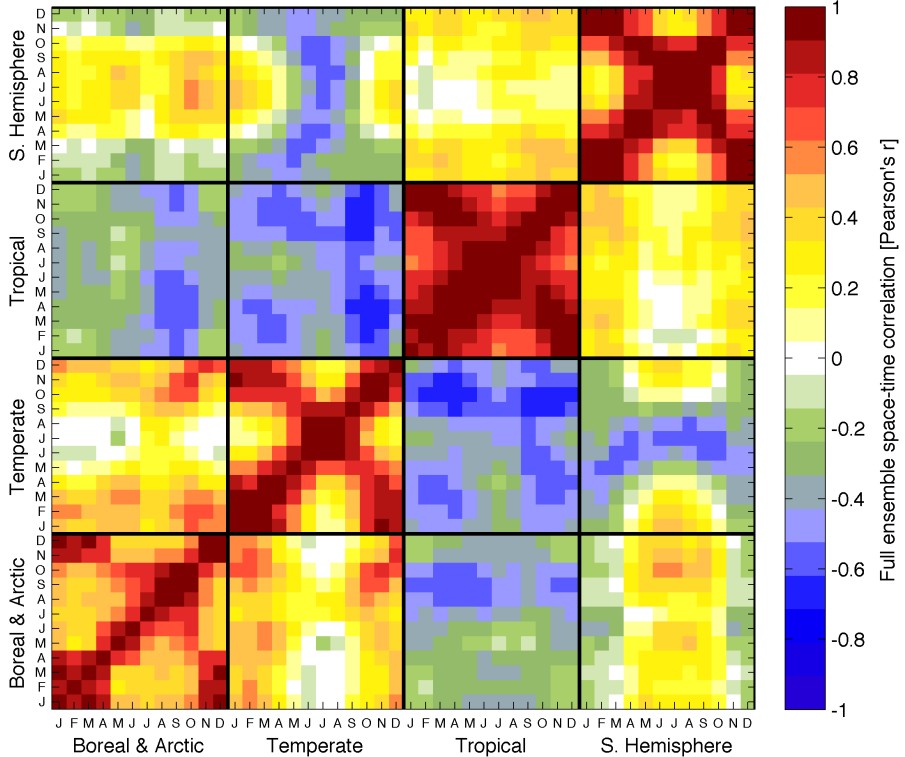

**Figure 6.** Full ensemble (FE) spatial and temporal error covariance, summarized as monthly error correlation across boreal & arctic (>55°N) temperate (23°N – 55°N), tropical (23°S – 23°N) and southern hemisphere (<23°S) latitudes. A correlation between two location-and-time indices indicates

5    the degree to which models consistently over- or under-predict wetland $CH_4$ emissions relative to the ensemble mean. The non-zero off-diagonal correlation patterns emerge as a function of varying biogeochemical commonalities across ensemble members, such as wetland $CH_4$ dependencies on temperature, carbon availability and wetland extent. Negative correlations between tropical and northern hemisphere extratropical (i.e. temperate, boreal and arctic) wetlands emerge as a function of a

10    global constraint on wetland $CH_4$ emissions (175 Tg $CH_4$ yr$^{-1}$ ± 19%).



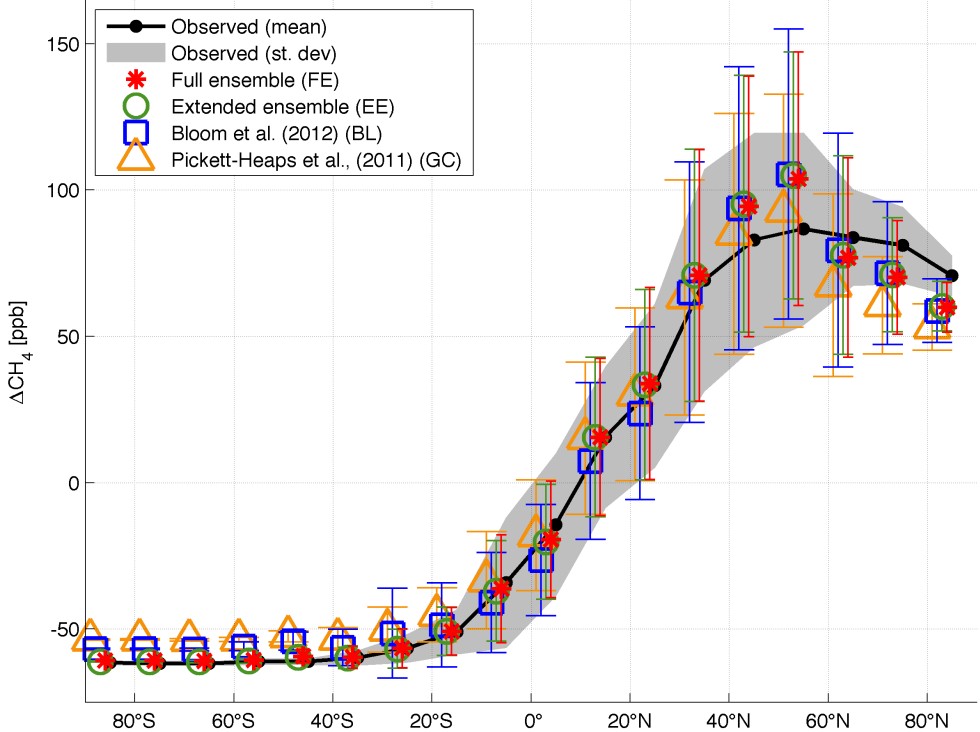

**Figure 7.** Mean 2009-2010 CH$_4$ measurements and model CH$_4$ zonal anomalies (ΔCH$_4$), relative to the mean 2009-2010 global CH$_4$ concentration. The black dots denote mean WDCGG network observed CH$_4$ concentrations within 5° latitude bins; the grey envelope denotes the mean 2009-2010 standard deviation across all sites within 5° latitude bins. The coloured symbols and error bars denote the GEOS-Chem equivalent model concentrations statistics based on the FE and EE ensembles (this study), Bloom et al., (2012) (BL) and the GEOS-Chem emission inventory (GC) wetland CH$_4$ emissions datasets.



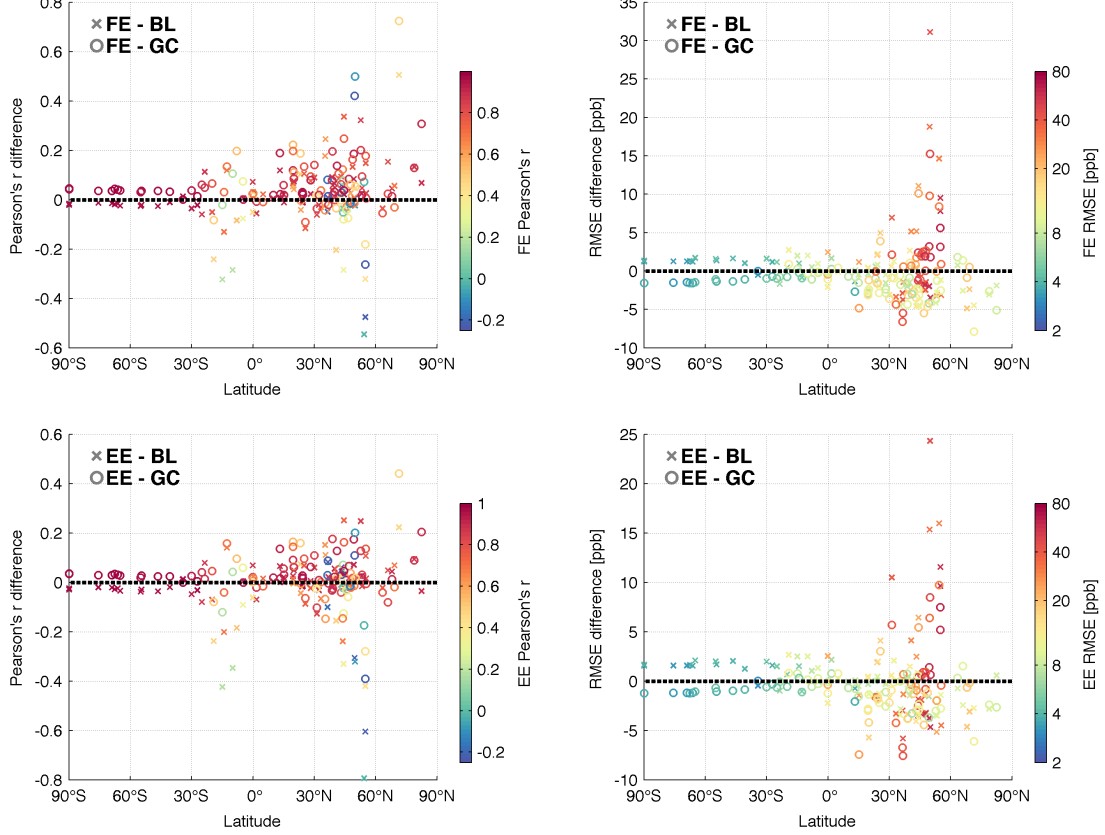

**Figure 8.** Symbol colours denote the monthly de-trended CH$_4$ model-observation Pearson's r correlation (left column) and RMSE (right column) for the FE (top-row) and EE (bottom-row) wetland CH$_4$ emissions (CH$_4$ observations are from the WDCGG measurement site network). The y-axis
5  denotes the difference between FE/EE and model runs with Bloom et al., (2012) wetland CH$_4$ emissions (BL) and the GEOS-Chem wetland CH$_4$ emissions inventory (GC).



**Tables**

**Table 1:** Wetland $CH_4$ model ensemble configurations

| Parameter | Description | | Ensemble configurations |
|---|---|---|---|
| $s$ | Global scaling factor[*] | | N/A (all configurations are scaled to 175 Tg $CH_4$ $yr^{-1}$) |
| $A$ | Wetland extent | *Spatial Extent* | 2 spatial extent parameterization (scaled using GLOBCOVER and GLWD) |
| | | *Temporal Variability* | SWAMPS inundation extent[**] |
| | | | ERA-interim precipitation |
| $R$ | Heterotrophic respiration | | 8 MsTMIP terrestrial C models[**] |
| | | | CARDAMOM terrestrial C cycle analysis |
| $q_{10(c)}$ | Temperature-dependent $CH_4$ respiration fraction. | | 3 $CH_4$:C temperature parameterizations: $q_{10(c)} = [1,2,3]$ |

[*]Global scaling factor is derived such that the global annual $CH_4$ emissions amount to 175 Tg $CH_4$ $yr^{-1}$.

5  [**] These datasets are only used in the 2009-2010 "Full Ensemble" (FE).