# Peer review of "A global wetland methane emissions and uncertainty dataset for atmospheric chemical transport models (WetCHARTs version 1.0)."

_Geoscientific Model Development, 2016_

## Referee Comment (RC1) · Anonymous Referee #1 · 30 Sep 2016

The authors aim at providing a dataset for methane emissions by wetlands which includes not only estimates of fluxes by biogeochemical models (the bottom-up approach) but also information on error covariance patterns. This information may be useful for performing inversions of methane fluxes through atmospheric data assimilation (i.e. for the top-down approach).

[Figure]

**General comments**

To my knowledge, this is the first time it is explicitly attempted to provide information on the uncertainty patterns together with bottom-up estimates of methane fluxes. As I am working with atmospheric data assimilation, I think the method and results of this study are very interesting. I have nevertheless two main remarks:

- the 6-member ensemble is too small to allow for statistics, so I would recommend only mentioning that "more classical" (i.e without the uncertainty patterns) methane flux estimates are available in EE for those who need a long period of time - and simplifying the text and figures accordingly in the Results and Discussion Sections

- the figures are potentially very nice and informative but presently difficult to read, even with a large zoom on a screen (see below for more specific comments on each).

**Specific comments**

General

- The means of the ensembles are used. Why not use the median?

- See if it is possible to update your *Kirschke et al.* (2013) reference with *Saunois et al.* (2016) (available at `http://www.earth-syst-sci-data-discuss.net/essd-2016-25/`) in the whole text. In Section 2.1, p.7, l.17-18: would propagating the new smaller uncertainty for global mean wetland methane emissions *(i)* take much time and *(ii)* significantly change the results?

Section 1 Introduction

- p.3, l.20-21: uncertainties are often formulated using correlation lengths in space (e.g. at the global scale, 500 km on land) and sometimes also in time (e.g, still at the global scale, one or two months) over a percentage of the prior emissions. This is especially done to take into account large patterns in the errors due to under-lying controls as is the case with wetland emissions. Please check your references here and adapt the text. This does not change the fact that correlation lengths are always an issue because the value at which they are set is derived from expert-knowledge, which is mainly valid at the global scale.

- p.4, l.8-11: I don't understand here what is meant by the "further constrained" ensemble. Do you mean that the top-down approach could retrieve the controls of biogeochemical processes instead of fluxes from atmospheric data assimilation? It seems it is what is meant in Section 4.2, p.15, l.6-8.

Section 2.1

- p.5, l.20: using the word "ensemble" for a six-member sub-set and deriving statistics over such a small number of members does not seem very appropriate. See if it is possible to leave EE aside for most of the paper and only mention it as a "more classical" set of estimates (see also General comments).

- p.7, l.18-19: why 1000 perturbations?

Section 2.2

- p.9, l.4-6: I understand the idea of keeping mostly sites where the vertical mixing in the model is not too much of an issue but using only the altitude (a.s.l.?) of the site seems to be too simple. Could you detail a bit more?

[Figure]

Section 3

- see if it is possible to leave EE aside (see above and General comments)

- p.10, l.5: could you quantify "considerable"?

- p.10, l.19-21: could you explain in more detail why you expect inter-annual variability to be smaller than your uncertainty?

- p.11, l.16-seq.: this paragraph is difficult to read with all the figures embedded in the text. Could you put them in a Figure or Table?

Section 4.1

- p.14, l.8-11: more and more atmospheric data of mixing ratios of methane isotopes are available and data assimilation systems try to make use of these and isotopic signatures of the various sources to improve the inversion of methane fluxes. Do you think not only the total methane fluxes but also the isotopic composition could be improved?

- p.14, l.16: the global uncertainty is always smaller than the smaller scale uncertainties, could you quantify "substantially"?

**Technical corrections**

General

- check "Kirschke" everywhere (and not "Kirshke")

- check all references in the form of "based on the *Bloom et al.*, (2016) methodology": it looks like there shouldn't be a comma before the year between parenthesis.

- "primary" is used for "main" or "dominant" e.g. p.7, l.26 or p.11, l.20, and it seems a bit strange to me, non-native English speaker.

Section 1 Introduction

- p.4, l.9: "based top-down $CH_4$ emission estimates" → based ON top-down $CH_4$ emission estimates?

Section 2.1

- p.5, l.10: "heterotrophic respiration at time for a" → delete "at time"?

- p.6, l.18: **h** should be *h*?

- p.7, l.4: "freshwater bodies in $w_i(x)$ section 4" → delete $w_i(x)$?

Section 3

- p.9, l.18: if FE emissions are intended, it seems that it should be Figure 1a; if it is Figure 1b which is commented, it should be "High-latitude EE emissions".

- p.9, l.19-20: add references to panels c and d in Figure 1 to guide the reader.

- p.9, l.22: "(EE) s; the FE" → delete "s"?

- p.10, l.14: Chang et al. (2014) should be "Alaska Wetlands" to be consistent with the whole sentence.

- p.10, l.24-25: "dominated carbon decomposition" → "dominated BY carbon de-composition"

- p.13, l.8: "is able capture" -> "is able TO capture"

Figures

- Figure 1: difficult to read, even on a screen. Larger maps and discrete colour scales would make it easier. It seems that panels e and f are never referred to in the text.

- Figure 2:

  – it is almost impossible to distinguish pale grey fine lines from darker grey larger lines!
  – put FE ensemble and FE mean in the legend since it is used in the whole text (instead of "Ensemble" / "Mean" alone in the top panel or "This study" in the bottom panel)
  – if following the recommendation of not commenting too much on EE in the body of the article, the top panel could be in Supplementary material (or Appendix?)

- Figure 3: a discrete colour scale would make it easier to read, together with larger panels if possible.

- Figure 4: you may use box plots to make the legend clearer and shorter; could you enlarge the map?

- Figure 6: the colour scale is a bit strange since the ticks every 0.2 do not fit the limits of shades

- Figure 8: a discrete colour scale would make it easier to read

Appendix

- Appendix B:
  - p.16, l.10: "ensemble. the "cor()"" → "ensemble. The "cor()""
  - p.16, l.11: "For Figure 6, $A_{l,m}$ we aggregate": do you mean that Fig 6 shows the $A_{l,m}$ coefficients?

- Appendix C: p.17, l.4: "$R_{100,1,1}$" should probably be $R_{100,3,1}$

---

## Referee Comment (RC2) · Anonymous Referee #2 · 4 Oct 2016

This study describes and evaluates a new global dataset of CH4 emissions from natural wetlands. The method follows an ensemble approach, which has the advantage that the computation of uncertainties, including spatio-temporal covariances, is straightforward. The dataset as meant to serve as a first guess in inverse modeling for which the uncertainty quantification has a clear advantage over other methods. It is not entirely clear what the evaluation using the GEOS-CHEM model brings, other than the notion that this dataset is in reasonable agreement with datasets that were used in the past. Obviously, flux measurements are better suited to test the performance of a methane emission model, although scale dependencies complicate that approach also. Since this holds for the other datasets as well, it would nevertheless provide additional information. Otherwise I was missing the dimension of inter-annual variability, which brings a clear advantage for the EE dataset - although it remains unclear what that variation looks like and how realistic it is. Otherwise I have only a list of technical corrections, which should be relatively easy to tackle.

page 4, line 9: 'based top down'

page 5, eq. 20: Mention that there are 6 scenarios for EE (which helps the reader to make sure he/she understands table 1 correctly)

eq 2: what is done when w(x) is not covered by h(x) and vice verse?

eq 3: how is this done for the EE time series, every year 175Tg/yr or just the mean over the whole period? In the latter case: how do the global emissions compare for 2009-2010? It would also be useful to know how much of a correction is needed to get to 175 Tg/yr.

page 7, line 26: 'uncertainty. The derivation' i.o. 'uncertainty; the derivation'

page 8, line 11: 'been in a'

page 8, line 15: 'Commission' i.o. 'Comission'

page 8, line 17: '). The non-wetland' i.o. ') .The non-wetland'

page 9, line 23: 'significantly lower (with' i.o. 'significantly (with' and remove 'lower' in the next line.

page 10, line 20: 'estimated' i.o. 'estimate'

page 10, line 25: 'by carbon' i.o. 'carbon'

page 11, line 15 - bottom: This part is hard to read due to all the numbers. It would be better to put the numbers in a Table.

page 13-14: How much emissions are derived from rivers/lakes using the current approach?

page 28: 'Contribution' i.o. 'Contri- bution'

figure 1: bottom panels: how can the units be compared?

figure 3: the legend title misses a unit area.

————————————————

---

## Short Comment (SC1) · 24 Oct 2016

Dear authors,

In my role as Executive editor of GMD, I would like to bring to your attention our Editorial version 1.1:

http://www.geosci-model-dev.net/8/3487/2015/gmd-8-3487-2015.html

This highlights some requirements of papers published in GMD, which is also available on the GMD website in the 'Manuscript Types' section:

http://www.geoscientific-model-development.net/submission/manuscript_types.html

[Figure]

In particular, please note that for your paper, the following requirements have not been met in the Discussions paper:

- "Inclusion of Code and/or data availability sections is mandatory for all papers and should be located at the end of the article, after the conclusions, and before any appendices or acknowledgments. For more details refer to the code and data policy" (Editorial v1.1, Appendix A1)

- "Papers describing data sets designed for the support and evaluation of model simulations are within scope. These data sets may be syntheses of data which have been published elsewhere. The data sets must also be made available, and any code used to create the syntheses should also be made available." (Editorial v1.1, Appendix A5)
  For these papers the same criteria as for model description papers apply, i.e., "The main paper must give the model name and version number (or other unique identifier) in the title." (Editorial v1.1, Appendix A2)

  In this case the "model" is the "data set".

Please add a data availability section and include the data sets name and version number in the title in your revised submission to GMD.

Yours,

Astrid Kerkweg

---

## Author Comment (AC1) · 28 Feb 2017

**We thank the reviewers for their constructive feedback and suggested corrections. Below we have addressed each individual comment from reviewers 1 and 2, as well as comments by executive editor Astrid Kerkweg (reviewer and editor comments are shown in italics; our responses to the reviewer comments are shown in bold). We believe that the following revisions have substantially improved the overall quality of our manuscript.**

*Referee 1*

*The authors aim at providing a dataset for methane emissions by wetlands which*

*includes not only estimates of fluxes by biogeochemical models (the bottom-up approach) but also information on error covariance patterns. This information may be useful for performing inversions of methane fluxes through atmospheric data assimilation (i.e. for the top-down approach).*

*General comments*

*To my knowledge, this is the first time it is explicitly attempted to provide information on the uncertainty patterns together with bottom-up estimates of methane fluxes. As I am working with atmospheric data assimilation, I think the method and results of this study are very interesting. I have nevertheless two main remarks:*

**(1.1)** *the 6-member ensemble is too small to allow for statistics, so I would recommend only mentioning that "more classical" (i.e without the uncertainty patterns) methane flux estimates are available in EE for those who need a long period of time - and simplifying the text and figures accordingly in the Results and Discussion Sections*

**We acknowledge that relative to FE, the EE ensemble size may limit statistical representations of wetland CH$_4$ model uncertainty. We now elaborate on this point in the discussion section of the revised manuscript: " due to the smaller ensemble size and the use of only one carbon model (see Table 1), the 2001-2015 EE emission variability should be interpreted with caution, and - where possible - evaluated against the FE ensemble during the 2009-2010 period".**

**However, we have chosen to keep the EE evaluations, as these may be beneficial for the users of EE. We also note that - in response to comment 1.8 - we have expanded the EE ensemble (EE ensemble size = 18) to explicitly represent the uncertainty of the global wetland CH$_4$ source. Finally, at the request of the second reviewer, we have now included an evaluation of the EE emission inter-annual variability in the revised manuscript. Given these changes, we are confident that the inclusion of EE evaluations is fitting for the main body of the manuscript.**

**(1.2)** *The figures are potentially very nice and informative but presently difficult to read, even with a large zoom on a screen (see below for more specific comments on each).*

**We have addressed the reviewer's comments relating to figures below; where appropriate, we have also increased font and panel sizes to improve figure legibility.**

*Specific Comments*

*General*

**(1.3)** *The means of the ensembles are used. Why not use the median?*

**For gridded and zonal emission estimates (Figures 1-3 and GEOS-Chem simulations), we chose to report the mean values in order to maintain consistency with the prescribed global wetland CH$_4$ source (166 Tg CH$_4$ yr$^{-1}$ during 2009-2010; see comment 1.4). We found that gridded and zonal FE and EE median values amount to substantially less than 166 CH$_4$ yr$^{-1}$; in contrast, mean FE and EE emissions amount to exactly 166 CH$_4$ yr$^{-1}$ during 2009-2010.**

**(1.4)** *See if it is possible to update your Kirschke et al. (2013) reference with Saunois et al. (2016) (available at http://www.earth-syst-sci-data-discuss.net/essd-2016-25/) in the whole text. In Section 2.1, p.7, l.17-18: would propagating the new smaller uncertainty for global mean wetland methane emissions (i) take much time and (ii) significantly change the results?*

**As suggested by the reviewer, we have now updated the global mean wetland CH$_4$ emission estimate and the associated uncertainty (166 Tg/yr +/- 25% or 124.5 - 207.5 Tg/yr), which spans the Saunois et al., (2016) mean (166 Tg/yr) and range (125 - 204 Tg/yr) of 2000-2009 wetland CH$_4$ emission estimates. We have updated the text, figures and the GEOS-Chem model simulations accordingly. We did not find any substantial changes in the results or evaluation of our dataset.**

*Section 1 Introduction*

**(1.5)** *p.3, l.20-21: uncertainties are often formulated using correlation lengths in space (e.g. at the global scale, 500 km on land) and sometimes also in time (e.g, still at the global scale, one or two months) over a percentage of the prior emissions. This is especially done to take into account large patterns in the errors due to underlying controls as is the case with wetland emissions. Please check your references here and adapt the text. This does not change the fact that correlation lengths are always an issue because the value at which they are set is derived from expert-knowledge, which is mainly valid at the global scale.*

**We acknowledge our oversight: we omitted to mention the use of prior spatial and temporal correlations on total CH$_4$ emissions, such as those used in the Bousquet et al., (2011) and Pison et al., (2013) studies. We have rephrased the sentence to better summarize the use of prior error covariances among inversion efforts: "typically CH$_4$ inversions do not explicitly formulate wetland CH$_4$ emission uncertainty correlations: rather, prior wetland CH$_4$ uncertainty correlations are either absent or implicitly prescribed through space-time correlation lengths on total CH$_4$ emissions."**

**(1.6)** *p.4, l.8-11: I don't understand here what is meant by the "further constrained" ensemble. Do you mean that the top-down approach could retrieve the controls of biogeochemical processes instead of fluxes from atmospheric data assimilation? It seems it is what is meant in Section 4.2, p.15, l.6-8.*

**We have now re-worded this sentence: "Top-down CH$_4$ emission estimates can then be used to quantify (a) the probability of individual ensemble members; and (b) the combined probability distribution of carbon models, CH$_4$:C temperature dependencies and wetland extent scenarios."**

*Section 2.1*

**(1.7)** *p.5, l.20: using the word "ensemble" for a six-member sub-set and deriving statistics over such a small number of members does not seem very appropriate. See if it*

*is possible to leave EE aside for most of the paper and only mention it as a "more classical" set of estimates (see also General comments).*

**We have revised our manuscript text and figures to better convey the limitations of the extended ensemble (see response to comment 1.1). We also note that the EE ensemble is now comprised of 18 members (see response to comment 1.8).**

**(1.8)** *p.7, l.18-19: why 1000 perturbations?*

**In response to the reviewer's question, we investigated the impact of the number of perturbations, and found no substantial impact on reported FE percentile intervals and error correlation values. In light of this, we now use a simpler approach to explicitly represent global wetland source uncertainty: we expand both FE and EE emission ensembles by deriving three scaling factors corresponding to global annual wetland emissions of 124.5, 166 and 207.5 Tg/yr; these span the range of the Saunois et al., 2016 emission estimates. The number of ensemble members for FE and EE are now 324 and 18. We have amended Table 1 and the methods section accordingly, and we have removed the description and calculation of FE$_{exp}$, since it is now obsolete.**

*Section 2.2*

**(1.9)** *p.9, l.4-6: I understand the idea of keeping mostly sites where the vertical mixing in the model is not too much of an issue but using only the altitude (a.s.l.?) of the site seems to be too simple. Could you detail a bit more?*

**Our statement was erroneous, as we did not actually use an altitude threshold (the text was a remnant of a previous analysis); we have corrected the sentence accordingly.**

*Section 3*

**(1.10)** *see if it is possible to leave EE aside (see above and General comments)*

**See response to comment (1.1).**

**(1.11)** *p.10, l.5: could you quantify "considerable"?*

**We now explicitly state the peak CH$_4$ emission month for each dataset.**

**(1.12)** *p.10, l.19-21: could you explain in more detail why you expect inter-annual variability to be smaller than your uncertainty?*

**In the revised manuscript, we have revised this sentence to clarify that our expectation is based on previous wetland CH$_4$ modelling efforts. We also support our statement with a quantitative comparison between the FE and EE uncertainties and the maximum inter-annual variability of the 1993-2004 WETCHIMP models.**

**(1.13)** *p.11, l.16-seq.: this paragraph is difficult to read with all the figures embedded in the text. Could you put them in a Figure or Table?*

**In the revised manuscript, we have split the paragraph into two (as two separate figures are being presented), and we have omitted the 5th - 95th percentile results, as these are depicted graphically in Figure 8.**

*Section 4.1*

**(1.14)** *p.14, l.8-11: more and more atmospheric data of mixing ratios of methane isotopes are available and data assimilation systems try to make use of these and isotopic signatures of the various sources to improve the inversion of methane fluxes. Do you think not only the total methane fluxes but also the isotopic composition could be improved?*

**We agree that our ensemble estimates can be used to better represent the biogeochemical process uncertainty in isotopic CH$_4$ studies; however, we have chosen not describe the potential advantages of using our datasets in isotopic CH$_4$ investigations, as these are beyond the scope of our manuscript. In case the**

reviewer is specifically referring to lakes and wetlands: to the best of our knowledge, lake and wetland isotopic $CH_4$ signatures are not sufficiently distinct to resolve between the two sources.

**(1.15)** *p.14, l.16: the global uncertainty is always smaller than the smaller scale uncertainties, could you quantify "substantially"?*

**We have revised this sentence and now explicitly state that regional scale uncertainties (shown in Figure 4) span a factor of 2 – 148.**.

*Technical Corrections*

*General*

**(1.16)** *check "Kirschke" everywhere (and not "Kirshke")*

**Done**

**(1.17)** *check all references in the form of "based on the Bloom et al., (2016) methodology": it looks like there shouldn't be a comma before the year between parenthesis.*

**Done**

**(1.18)** *"primary" is used for "main" or "dominant" e.g. p.7, l.26 or p.11, l.20, and it seems a bit strange to me, non-native English speaker.*

**In reference to the "primary uncertainty" estimates, we have now replaced "primary" by "dominant" throughout the revised manuscript.**

*Section 1 Introduction*

**(1.19)** *p.4, l.9: "based top-down CH4 emission estimates" → based ON top-down CH4 emission estimates?*

**This sentence has been edited in the revised manuscript**

*Section 2.1*

[Figure]

**(1.20)** *p.5, l.10: "heterotrophic respiration at time for a" → delete "at time"?*

**Now changed to "heterotrophic respiration per unit area at time t"**

**(1.21)** *p.6, l.18: **h** should be h?*

**h is now written in bold-italics as "$\boldsymbol{h}_{*,j}$" is a vector. We note that in the revised manuscript mathematical notations are now consistent with the Geophysical Model Development journal requirements.**

**(1.22)** *p.7, l.4: "freshwater bodies in wi(x) section 4" → delete wi(x)?*

**Typo corrected**

*Section 3*

**(1.23)** *p.9, l.18: if FE emissions are intended, it seems that it should be Figure 1a; if it is Figure 1b which is commented, it should be "High-latitude EE emissions".*

**In the revised manuscript we have re-worded this paragraph in order to (a) correctly reference the spatial distributions of FE and EE emissions; (b) add explicit references to panels c and d (as recommended by the reviewer in comment 1.24).**

**(1.24)** *p.9, l.19-20: add references to panels c and d in Figure 1 to guide the reader.*

**Done**

**(1.25)** *p.9, l.22: "(EE) s; the FE" → delete "s"?*

**Typo corrected**

**(1.26)** *p.10, l.14: Chang et al. (2014) should be "Alaska Wetlands" to be consistent with the whole sentence.*

**Done**

**(1.27)** *p.10, l.24-25: "dominated carbon decomposition" → "dominated BY carbon decomposition"*

**Done**

**(1.28)** *p.13, l.8: "is able capture" → "is able TO capture"*

**Done**

*Figures*

**(1.29)** *Figure 1: difficult to read, even on a screen. Larger maps and discrete colour scales would make it easier. It seems that panels e and f are never referred to in the text.*

**As recommended by the reviewer, we have now enlarged the fontsizes, increased the map sizes, and have used a discrete (9-color) scale. We have also included explicit references to panels e and f in the results section of the revised manuscript.**

**(1.30)** *Figure 2: - it is almost impossible to distinguish pale grey fine lines from darker grey larger lines! - put FE ensemble and FE mean in the legend since it is used in the whole text (instead of "Ensemble" / "Mean" alone in the top panel or "This study" in the bottom panel) - if following the recommendation of not commenting too much on EE in the body of the article, the top panel could be in Supplementary material (or Appendix?)*

**For the sake of clarity, we have now removed the top panel in Figure 2; we now show the means and full ranges of FE and EE emissions in a single panel along with GC and BL emissions. We have updated the figure legend accordingly.**

**(1.31)** *Figure 3: a discrete colour scale would make it easier to read, together with larger panels if possible.*

**We have now revised the figure to include a discrete color scale; as recommended, we have also expanded the panel sizes**.

**(1.32)** *Figure 4: you may use box plots to make the legend clearer and shorter; could*

*you enlarge the map?*

**As recommended, we have substantially enlarged the map in the revised figure and shortened the legend.**

**(1.33)** *Figure 6: the colour scale is a bit strange since the ticks every 0.2 do not fit the limits of shades*

**We have revised the color scale to match the ticks.**

**(1.34)** *Figure 8: a discrete colour scale would make it easier to read C6*

**The revised figure includes a discrete color scheme.**

**(1.35)** *Appendix B: - p.16, l.10: "ensemble. the "cor()"" → "ensemble. The "cor()""*

**Typo corrected**

*- p.16, l.11: "For Figure 6, Al,m we aggregate": do you mean that Fig 6 shows the Al,m coefficients?*

**Sentence revised: the start of this sentence now reads "For Figure 6, we aggregated..."**

**(1.36)** *Appendix C: p.17, l.4: "R100,1,1" should probably be R100,3,1*

**Typo corrected**

*Anonymous Referee 2*

*This study describes and evaluates a new global dataset of CH4 emissions from natural wetlands. The method follows an ensemble approach, which has the advantage that the computation of uncertainties, including spatio-temporal covariances, is straightforward. The dataset as meant to serve as a first guess in inverse modeling for which the uncertainty quantification has a clear advantage over other methods.*

**(2.1)** *It is not entirely clear what the evaluation using the GEOS-CHEM model brings,*

*other than the notion that this dataset is in reasonable agreement with datasets that were used in the past. Obviously, flux measurements are better suited to test the performance of a methane emission model, although scale dependencies complicate that approach also. Since this holds for the other datasets as well, it would nevertheless provide additional information.*

**We agree with the reviewer that - relative to the regional flux constraints (Figure 4) - the GEOS-Chem evaluation only provides supporting evidence on the plausibility of wetland CH$_4$ emissions relative to previous datasets: we now clarify this point in the revised manuscript. In addition, we now recognize the value of in-situ methane CH$_4$ measurements in the revised manuscript. While the direct comparison between global-scale fluxes and in-situ measurements is a challenging task (and beyond the scope of our work) we now highlight the importance of measurement-based regional estimates for the evaluation of global wetland CH4 emission ensembles.**.

**(2.2)** *Otherwise I was missing the dimension of inter-annual variability, which brings a clear advantage for the EE dataset - although it remains unclear what that variation looks like and how realistic it is.*

**We have now incorporated an evaluation of the EE inter-annual variability (IAV), including an additional manuscript figure: in particular, we compare 2001-2015 EE IAV against 2009-2010 FE emissions and the WETCHIMP model ensemble during 2001-2004. We also include an evaluation EE IAV against observationally constrained Alaska CH$_4$ emissions during 2012-2014 (Miller et al., 2016) and Amazon emissions during 2010-2011 (Wilson et al., 2016). Our evaluation indicates that EE IAV is broadly consistent with both modeled and observationally constrained wetland CH4 emission estimates.**

*Otherwise I have only a list of technical corrections, which should be relatively easy to tackle.*

**(2.3)** *page 4, line 9: 'based top down'*

**This sentence has been edited in the revised manuscript**

**(2.4)** *page 5, eq. 20: Mention that there are 6 scenarios for EE (which helps the reader to make sure he/she understands table 1 correctly)*

**Done: we note that EE now consists of 18 ensemble members (see response to comment 1.8)**

**(2.5)** *eq 2: what is done when w(x) is not covered by h(x) and vice verse?*

**We have revised the description of equation 2 to clarify that w(x) represents the wetland extent fraction, while h(x,t) is the relative temporal variability. In the revised manuscript, we also clarify that when h(x,t)w(x) > 1, A(x,t) is set to a maximum value of 1.**

**(2.6)** *eq 3: how is this done for the EE time series, every year 175Tg/yr or just the mean over the whole period? In the latter case: how do the global emissions compare for 2009-2010? It would also be useful to know how much of a correction is needed to get to 175 Tg/yr.*

**For both EE and FE, $s_c$ is derived such that each ensemble member's global emissions amount to an average annual flux of 124.5, 166 or 207.5 Tg/yr during the 2009-2010 time period (previously 175 Tg/yr; see comment 1.8). We have now clarified this in the revised manuscript. In response to comment 2.2, we also include an evaluation of EE throughout 2001-2015. For the sake of brevity, individual FE and EE $s_c$ values (spanning 4 - 100% relative to the maximum $s_c$ value) will be included as part of the final manuscript dataset (the dataset will be linked to the manuscript via a digital object identifier; see comment 3.1).**

**(2.7)** *page 7, line 26: 'uncertainty. The derivation' i.o. 'uncertainty; the derivation'*

**Done**

**(2.8)** *page 8, line 11: 'been in a'*

**Done**

**(2.9)** *page 8, line 15: 'Commission' i.o. 'Comission'*

**Done**

**(2.10)** *page 8, line 17: '). The non-wetland' i.o. ') .The non-wetland'*

**Done**

**(2.11)** *page 9, line 23: 'significantly lower (with' i.o. 'significantly (with' and remove 'lower' in the next line.*

**Done**

**(2.12)** *page 10, line 20: 'estimated' i.o. 'estimate'*

**Changed to "emission uncertainty estimates are"**

**(2.12)** *page 10, line 25: 'by carbon' i.o. 'carbon'*

**Done**

**(2.13)** *page 11, line 15 - bottom: This part is hard to read due to all the numbers. It would be better to put the numbers in a Table.*

**See response to reviewer comment (1.13)**

**(2.14)** *page 13-14: How much emissions are derived from rivers/lakes using the current approach?*

**As discussed in the "model limitations" section, we are unable to report emissions from rivers and lakes as we have insufficient information to disentangle the relative CH$_4$ contribution of non-wetland freshwater bodies within each grid-cell. We have re-worded this section to clarify this point in the revised manuscript.**

[Figure]

**(2.15)** *page 28: 'Contribution' i.o. 'Contri- bution'*

**Done**

**(2.16)** *figure 1: bottom panels: how can the units be compared?*

**All color bar labels now include units in the revised version of Figure 1**.

**(2.17)** *figure 3: the legend title misses a unit area*

textbfThe wetland emissions (Tg CH$_4$ yr$^{-1}$) are totals within each region shown on the inset map. We have modified the figure caption to clarify this.

*Astrid Kerkweg*

**(3.1)** *In my role as Executive editor of GMD, I would like to bring to your attention our Editorial version 1.1: http://www.geosci-model-dev.net/8/3487/2015/gmd-8-3487-2015.html. This highlights some requirements of papers published in GMD, which is also available on the GMD website in the 'Manuscript Types' section: http://www.geoscientific-model-development.net/submission/manuscript_types.html. In particular, please note that for your paper, the following requirements have not been met in the Discussions paper:*

*"Inclusion of Code and/or data availability sections is mandatory for all papers and should be located at the end of the article, after the conclusions, and before any appendices or acknowledgments. For more details refer to the code and data policy" (Editorial v1.1, Appendix A1)*

*"Papers describing data sets designed for the support and evaluation of model simulations are within scope. These data sets may be syntheses of data which have been published elsewhere. The data sets must also be made available, and any code used to create the syntheses should also be made available." (Editorial v1.1, Appendix A5).*

*For these papers the same criteria as for model description papers apply, i.e., "The main paper must give the model name and version number (or other unique identifier)*

*in the title." (Editorial v1.1, Appendix A2) In this case the "model" is the "data set"*

*Please add a data availability section and include the data sets name and version number in the title in your revised submission to GMD.*

**The data availability section is now positioned at the end of the article before the acknowledgments and appendices sections. We have now added a model name and version number (WetCHARTs version 1.0) in the title of our manuscript. We have also added a data availability section in the revised manuscript, and we will be making the code available in the supplementary material. The final dataset (>50 MB) will be submitted to the Oak Ridge National Laboratory Distributed Active Archive Center (ORNL DAAC), and will be linked to the manuscript via a digital object identifier (doi). The data availability section has been expanded to include information on the ancillary datasets used in this study.**

**Additional changes**

**In the revised manuscript, Figure 1e now correctly shows the FE 5th - 95th percentile values (in the discussion manuscript, the figure was inadvertently showing FE standard deviation values).**

**References not included in the discussion manuscript:**

**Miller, Scot M., et al.: A multiyear estimate of methane fluxes in Alaska from CARVE atmospheric observations. Global Biogeochemical Cycles 30 (10) 1441-1453, 2016**

**Saunois, M., et al.: The global methane budget 2000-2012, Earth Syst. Sci. Data, 8, 697-751, 2016.**
* * *

---

## Author Response (AR1)

We thank the reviewers for their constructive feedback and suggested corrections. Below we have addressed each individual comment from reviewers 1 and 2, as well as comments by the executive editor (reviewer and editor comments are shown in italics; our responses are shown in bold). All manuscript changes are highlighted as 'tracked changes' in the revised manuscript (the bracketed line numbers denote the corresponding line numbers in the revised manuscript). We believe that the following revisions have substantially improved the overall quality of our manuscript.

**Referee 1**

The authors aim at providing a dataset for methane emissions by wetlands which includes not only estimates of fluxes by biogeochemical models (the bottom-up approach) but also information on error covariance patterns. This information may be useful for performing inversions of methane fluxes through atmospheric data assimilation (i.e. for the top-down approach).

**General comments**

To my knowledge, this is the first time it is explicitly attempted to provide information on the uncertainty patterns together with bottom-up estimates of methane fluxes. As I am working with atmospheric data assimilation, I think the method and results of this study are very interesting. I have nevertheless two main remarks:

(1.1) the 6-member ensemble is too small to allow for statistics, so I would recommend only mentioning that "more classical" (i.e without the uncertainty patterns) methane flux estimates are available in EE for those who need a long period of time - and simplifying the text and figures accordingly in the Results and Discussion Sections

We acknowledge that relative to FE, the EE ensemble size may limit statistical representations of wetland  $CH_4$  model uncertainty. We now elaborate on this point in the discussion section of the revised manuscript: "due to the smaller ensemble size and the use of only one carbon model (see Table 1), the 2001-2015 EE emission variability should be interpreted with caution, and - where possible - evaluated against the FE ensemble during the 2009-2010 period" (P16 L3-6).

However, we have chosen to keep the EE evaluations, as these may be beneficial for the users of EE. We also note that - in response to comment 1.8 - we have expanded the EE ensemble (EE ensemble size = 18) to explicitly represent the uncertainty of the global wetland  $CH_4$  source. Finally, at the request of the second reviewer, we have now included an evaluation of the EE emission inter-annual variability in the

revised manuscript (see response to comment 2.2). Given these changes, we are confident that the inclusion of EE evaluations is fitting for the main body of the manuscript.

(1.2) The figures are potentially very nice and informative but presently difficult to read, even with a large zoom on a screen (see below for more specific comments on each).

We have addressed the reviewer's comments relating to figures below; where appropriate, we have also increased font and panel sizes to improve figure legibility.

Specific Comments

General

(1.3) The means of the ensembles are used. Why not use the median?

For gridded and zonal emission estimates (Figures 1-3 and GEOS-Chem simulations), we chose to report the mean values in order to maintain consistency with the prescribed global wetland  $CH_4$  source (166 Tg  $CH_4$  yr-1 during 2009-2010; see comment 1.4). We found that gridded and zonal FE and EE median values amount to substantially less than 166  $CH_4$  yr-1; in contrast, mean FE and EE emissions amount to exactly 166  $CH_4$  yr-1 during 2009-2010.

(1.4) See if it is possible to update your Kirschke et al. (2013) reference with Saunois et al. (2016) (available at http://www.earth-syst-sci-data-discuss.net/essd-2016-25/) in the whole text. In Section 2.1, p.7, l.17-18: would propagating the new smaller uncertainty for global mean wetland methane emissions (i) take much time and (ii) significantly change the results?

As suggested by the reviewer, we have now updated the global mean wetland  $CH_4$  emission estimate and the associated uncertainty (166 Tg/yr +/- 25% or 124.5 - 207.5 Tg/yr), which spans the Saunois et al., (2016) mean (166 Tg/yr) and range (125 - 204 Tg/yr) of 2000-2009 wetland  $CH_4$  emission estimates. We have updated the text (P5 L22-25; P7 L16-19; P15 L10-13), figures, Table 1 and the GEOS-Chem model simulations accordingly. We did not find any substantial changes in the results or evaluation of our dataset.

**Section 1 Introduction**

(1.5) p.3, l.20-21: uncertainties are often formulated using correlation lengths in

space (e.g. at the global scale, 500 km on land) and sometimes also in time (e.g, still at the global scale, one or two months) over a percentage of the prior emissions. This is especially done to take into account large patterns in the errors due to underlying controls as is the case with wetland emissions. Please check your references here and adapt the text. This does not change the fact that correlation lengths are always an issue because the value at which they are set is derived from expert-knowledge, which is mainly valid at the global scale.

We acknowledge our oversight: we omitted to mention the use of prior spatial and temporal correlations on total  $CH_4$  emissions, such as those used in the Bousquet et al., (2011) and Pison et al., (2013) studies. We have rephrased the sentence to better summarize the use of prior error covariances among inversion efforts: "typically  $CH_4$  inversions do not explicitly formulate wetland  $CH_4$  emission uncertainty correlations: rather, prior wetland  $CH_4$  uncertainty correlations are either absent or implicitly prescribed through space-time correlation lengths on total  $CH_4$  emissions" (P3 L21-23).

(1.6) p.4, l.8-11: I don't understand here what is meant by the "further constrained" ensemble. Do you mean that the top-down approach could retrieve the controls of biogeochemical processes instead of fluxes from atmospheric data assimilation? It seems it is what is meant in Section 4.2, p.15, l.6-8.

We have now re-worded this sentence: "Top-down  $CH_4$  emission estimates can then be used to quantify (a) the probability of individual ensemble members; and (b) the combined probability distribution of carbon models,  $CH_4$ :C temperature dependencies and wetland extent scenarios" (P4 L12-14).

Section 2.1

(1.7) p.5, l.20: using the word "ensemble" for a six-member sub-set and deriving statistics over such a small number of members does not seem very appropriate. See if it is possible to leave EE aside for most of the paper and only mention it as a "more classical" set of estimates (see also General comments).

We have revised our manuscript text and figures to better convey the limitations of the extended ensemble (see response to comment 1.1). We also note that the EE ensemble is now comprised of 18 members (see response to comment 1.8)

(1.8) p.7, l.18-19: why 1000 perturbations?

In response to the reviewer's question, we investigated the impact of the number of

perturbations, and found no substantial impact on reported FE percentile intervals and error correlation values. In light of this, we now use a simpler approach to explicitly represent global wetland source uncertainty: we expand both FE and EE emission ensembles by deriving three scaling factors corresponding to global annual wetland emissions of 124.5, 166 and 207.5 Tg/yr; these span the range of the Saunois et al., (2016) emission estimates. The number of ensemble members for FE and EE are now 324 and 18. We have amended Table 1 and the methods section (P5 L22-25; P7 L16-19), and we have removed the description and calculation of  $FE_{exp}$ , since it is now obsolete.

Section 2.2

(1.9) p.9, l.4-6: I understand the idea of keeping mostly sites where the vertical mixing in the model is not too much of an issue but using only the altitude (a.s.l.?) of the site seems to be too simple. Could you detail a bit more?

Our statement was erroneous, as we did not actually use an altitude threshold (the text was a remnant of a previous analysis); we have modified the sentence accordingly (P9 L1).

Section 3

(1.10) see if it is possible to leave EE aside (see above and General comments)

See response to comment (1.1).

(1.11) p.10, l.5: could you quantify "considerable"?

**We now explicitly state the peak CH4 emission month for each dataset (P10 L12-13).**

(1.12) *p.10*, *l.19-21*: could you explain in more detail why you expect inter-annual variability to be smaller than your uncertainty?

We have now revised this sentence to clarify that our expectation is based on previous wetland  $CH_4$  process modelling efforts. For the sake of clarity, we also support our statement with a quantitative comparison between the FE and EE uncertainties and the maximum inter-annual variability of the 1993-2004 WETCHIMP models (P10 L26 - P11 L4).

(1.13) *p.11*, *l.16-seq.: this paragraph is difficult to read with all the figures embedded in the text. Could you put them in a Figure or Table?*  In the revised manuscript, we have split the paragraph into two, as two separate figures are being presented (P12 L10-24), and we have omitted the 5th - 95th percentile results, as these are depicted graphically in Figure 8.

Section 4.1

(1.14) p.14, l.8-11: more and more atmospheric data of mixing ratios of methane isotopes are available and data assimilation systems try to make use of these and isotopic signatures of the various sources to improve the inversion of methane fluxes. Do you think not only the total methane fluxes but also the isotopic composition could be improved?

We agree that our ensemble estimates can be used to better represent the biogeochemical process uncertainty in isotopic  $CH_4$  studies; however, we have chosen not describe the potential advantages of using our datasets in isotopic  $CH_4$  investigations, as these are beyond the scope of our manuscript. In case the reviewer is specifically referring to lakes and wetlands: to the best of our knowledge, lake and wetland isotopic  $CH_4$  signatures are not sufficiently distinct to resolve between the two sources.

(1.15) *p.14*, *l.16*: the global uncertainty is always smaller than the smaller scale uncertainties, could you quantify "substantially"?

We have revised this sentence and now explicitly state that regional scale uncertainties (shown in Figure 4) span a factor of 2 – 156 (P15 L8).

Technical Corrections

General

(1.16) check "Kirschke" everywhere (and not "Kirshke")

**Done**

(1.17) check all references in the form of "based on the Bloom et al., (2016) methodology": it looks like there shouldn't be a comma before the year between parenthesis.

**Done**

(1.18) "primary" is used for "main" or "dominant" e.g. p.7, l.26 or p.11, l.20, and it seems a bit strange to me, non-native English speaker.

In reference to the "primary uncertainty" estimates, we have now replaced "primary" by "dominant" throughout the revised manuscript.

Section 1 Introduction

(1.19) p.4, l.9: "based top-down CH4 emission estimates"  $\rightarrow$  based ON top-down CH4 emission estimates?

This sentence has been edited in the revised manuscript.

Section 2.1

(1.20) p.5, l.10: "heterotrophic respiration at time for a"  $\rightarrow$  delete "at time"?

Now changed to "heterotrophic respiration per unit area at time t" (P5 L13).

(1.21) *p.6*, *l.18*: *h* should be *h*?

h is now written in **bold-italics** as " $h_{*,j}$ " is a vector. We note that in the revised manuscript mathematical notations are now consistent with the Geophysical Model Development journal requirements.

(1.22) p.7, l.4: "freshwater bodies in wi(x) section 4"  $\rightarrow$  delete wi(x)?

Typo corrected (P7 L8)

Section 3

(1.23) p.9, l.18: if FE emissions are intended, it seems that it should be Figure 1a; if it is Figure 1b which is commented, it should be "High-latitude EE emissions".

In the revised manuscript we have re-worded this paragraph in order to (a) correctly reference the spatial distributions of FE and EE emissions; (b) add explicit references to panels c and d, as recommended by the reviewer in comment 1.24 (P9 L21 - P10 L3).

(1.24) p.9, l.19-20: add references to panels c and d in Figure 1 to guide the reader.

Done (P9 L25)

(1.25) p.9, l.22: "(EE) s; the FE"  $\rightarrow$  delete "s"?

**Typo corrected**

(1.26) p.10, l.14: Chang et al. (2014) should be "Alaska Wetlands" to be consistent with the whole sentence.

**Done (P10 L20-21)**

(1.27) p.10, l.24-25: "dominated carbon decomposition"  $\rightarrow$  "dominated BY carbon de- composition"

Done (P11 L19)

(1.28) *p.13*, *l.8*: "is able capture"  $\rightarrow$  "is able TO capture"

Done (P13 L28)

Figures

(1.29) Figure 1: difficult to read, even on a screen. Larger maps and discrete colour scales would make it easier. It seems that panels e and f are never referred to in the text.

As recommended by the reviewer, we have now enlarged the fontsizes, increased the map sizes, and have used a discrete (9-color) scale. We have also included explicit references to panels e and f in the results section of the revised manuscript (P9 L16-19).

(1.30) Figure 2: - it is almost impossible to distinguish pale grey fine lines from darker grey larger lines! - put FE ensemble and FE mean in the legend since it is used in the whole text (instead of "Ensemble" / "Mean" alone in the top panel or "This study" in the bottom panel) - if following the recommendation of not commenting too much on EE in the body of the article, the top panel could be in Supplementary material (or Appendix?)

For the sake of clarity, we have now removed the top panel in Figure 2; we now show the means and full ranges of FE and EE emissions in a single panel along with GC and BL emissions. We have updated the figure legend accordingly.

(1.31) Figure 3: a discrete colour scale would make it easier to read, together with larger panels if possible.

We have now revised the figure to include a discrete color scale; as recommended, we have also expanded the panel sizes.

(1.32) *Figure 4: you may use box plots to make the legend clearer and shorter; could you enlarge the map?*

As recommended, we have substantially enlarged the map in the revised figure and shortened the legend.

(1.33) Figure 6: the colour scale is a bit strange since the ticks every 0.2 do not fit the limits of shades

We have revised the color scale to match the ticks (now Figure 7).

(1.34) Figure 8: a discrete colour scale would make it easier to read C6

The revised figure (now Figure 9) includes a discrete color scheme.

(1.35) Appendix B: - p.16, l.10: "ensemble. the "cor()""  $\rightarrow$  "ensemble. The "cor()""

**Typo corrected**

- p.16, l.11: "For Figure 6, Al, m we aggregate": do you mean that Fig 6 shows the Al, m coefficients?

Sentence revised: the start of this sentence now reads "For Figure 7, we aggregated..." (P17 L25).

(1.36) Appendix C: p.17, l.4: "R100,1,1" should probably be R100,3,1

**Typo corrected (P18 L18)**

Anonymous Referee 2

This study describes and evaluates a new global dataset of CH4 emissions from natural wetlands. The method follows an ensemble approach, which has the advantage that the computation of uncertainties, including spatio-temporal covariances, is straightforward. The dataset as meant to serve as a first guess in inverse modeling for which the uncertainty quantification has a clear advantage over other methods.

(2.1) It is not entirely clear what the evaluation using the GEOS-CHEM model brings, other than the notion that this dataset is in reasonable agreement with datasets that were used in the past. Obviously, flux measurements are better suited to test the performance of a methane emission model, although scale dependencies complicate that approach also. Since this holds for the other datasets as well, it would nevertheless provide additional information.

We agree with the reviewer that - relative to the regional flux constraints (Figure 4) - the GEOS-Chem evaluation only provides supporting evidence on the plausibility of wetland  $CH_4$  emissions relative to previous datasets: we now clarify this point in the revised manuscript (P9 L3-5). In addition, we now recognize the value of in-situ methane  $CH_4$  measurements in the revised manuscript: while the direct comparison between global-scale fluxes and in-situ measurements is a challenging task (and beyond the scope of our work), we now highlight the importance of measurement-based regional estimates for the evaluation and improvement of future global wetland CH4 emission ensembles (P15 L10-13).

(2.2) Otherwise I was missing the dimension of inter-annual variability, which brings a clear advantage for the EE dataset - although it remains unclear what that variation looks like and how realistic it is.

We have now incorporated an evaluation of the EE inter-annual variability (IAV), including an additional manuscript figure: in particular, we compare 2001-2015 EE IAV against 2009-2010 FE emissions and the WETCHIMP model ensemble during 2001-2004 (see Figure 5 and P11 L6-11). We also include an evaluation EE IAV against observationally constrained Alaska  $CH_4$  emissions during 2012-2014 and Amazon emissions during 2010-2011 (P11 L11-15). Our evaluation indicates that EE IAV is broadly consistent with both modeled and observationally constrained wetland CH4 emission estimates.

Otherwise I have only a list of technical corrections, which should be relatively easy to tackle.

(2.3) page 4, line 9: 'based top down'

This sentence has been edited in the revised manuscript

(2.4) page 5, eq. 20: Mention that there are 6 scenarios for EE (which helps the reader to make sure he/she understands table 1 correctly)

Done (P5 L24): we note that EE now consists of 18 ensemble members (see response to comment 1.8)

(2.5) eq 2: what is done when w(x) is not covered by h(x) and vice verse?

We have revised the description of equation 2 to clarify that w(x) represents the wetland extent fraction, while h(t,x) is the relative temporal variability (P6 L15-16). In the revised manuscript, we also clarify that when h(t,x)w(x) > 1, A(t,x) is set to a maximum value of 1 (P6 L22-24).

(2.6) eq 3: how is this done for the EE time series, every year 175Tg/yr or just the mean over the whole period? In the latter case: how do the global emissions compare for 2009-2010? It would also be useful to know how much of a correction is needed to get to 175 Tg/yr.

For both EE and FE,  $s_c$  is derived such that each ensemble member's global emissions amount to an average annual flux of 124.5, 166 or 207.5 Tg/yr during the 2009-2010 time period (previously 175 Tg/yr; see comment 1.8). We have now clarified this in the revised manuscript (P7 L16-17). In response to comment 2.2, we also include an evaluation of EE throughout 2001-2015 (P11 L6-11 and Figure 5). For the sake of brevity, individual FE and EE  $s_c$  values (spanning 4 - 100% relative to the maximum  $s_c$  value) are not reported in the main body of the manuscript; however, we highlight that individual  $s_c$  values can be derived using the WetCHARTs code (now included in the supplementary material).

(2.7) page 7, line 26: 'uncertainty. The derivation' i.o. 'uncertainty; the derivation'

Done (P8 L1).

(2.8) page 8, line 11: 'been in a'

Sentence revised (P8 L8)

(2.9) page 8, line 15: 'Commission' i.o. 'Comission'

Typo corrected (P8 L12).

(2.10) page 8, line 17: '). The non-wetland' i.o. ') . The non-wetland'

Done (P8 L13).

(2.11) page 9, line 23: 'significantly lower (with' i.o. 'significantly (with' and remove 'lower' in the next line.

**Done (P10 L1)**

(2.12) page 10, line 20: 'estimated' i.o. 'estimate'

Changed to "emission uncertainty estimates are" (P10 L23).

(2.12) page 10, line 25: 'by carbon' i.o. 'carbon'

Done (P11 L19)

(2.13) page 11, line 15 - bottom: This part is hard to read due to all the numbers. It would be better to put the numbers in a Table.

See response to reviewer comment 1.13

(2.14) page 13-14: How much emissions are derived from rivers/lakes using the current approach?

As discussed in the "model limitations" section, we are unable to report emissions from rivers and lakes as we have insufficient information to disentangle the relative  $CH_4$  contribution of non-wetland freshwater bodies within each grid-cell. We have re-worded this section to clarify this point in the revised manuscript (P14 L24 - P15 L1).

(2.15) page 28: 'Contribution' i.o. 'Contri- bution'

Done

(2.16) figure 1: bottom panels: how can the units be compared?

All color bar labels now include units in the revised version of Figure 1.

(2.17) figure 3: the legend title misses a unit area

Wetland  $CH_4$  emissions (Tg/month) are reported as total emissions across 5-degree bins within each region shown on the inset map. We have modified the figure caption to clarify this.

Astrid Kerkweg

(3.1) In my role as Executive editor of GMD, I would like to bring to your attention our Editorial version 1.1: http://www.geosci-model-dev.net/8/3487/2015/gmd-8-3487-2015.html. This highlights some requirements of papers published in GMD, which is also available on the GMD website in the 'Manuscript Types' section: http://www.geoscientific-model-development.net/submission/manuscript\_types.html. In particular, please note that for your paper, the following requirements have not been met in the Discussions paper:

"Inclusion of Code and/or data availability sections is mandatory for all papers and should be located at the end of the article, after the conclusions, and before any appendices or acknowledgments. For more details refer to the code and data policy" (Editorial v1.1, Appendix A1)

"Papers describing data sets designed for the support and evaluation of model simulations are within scope. These data sets may be syntheses of data which have been published elsewhere. The data sets must also be made available, and any code used to create the syntheses should also be made available." (Editorial v1.1, Appendix A5).

For these papers the same criteria as for model description papers apply, i.e., "The main paper must give the model name and version number (or other unique identifier) in the title." (Editorial v1.1, Appendix A2) In this case the "model" is the "data set"

Please add a data availability section and include the data sets name and version number in the title in your revised submission to GMD.

The data availability section is now positioned at the end of the article before the acknowledgments and appendices sections (P16 L21). We have now added a model name and version number (WetCHARTs version 1.0) in the title of our manuscript. We have included the WetCHARTs matlab code in the supplementary material of our revised manuscript. The final dataset (>50 MB) has been submitted to the Oak Ridge National Laboratory Distributed Active Archive Center (ORNL DAAC): the dataset is linked to the manuscript via a digital object identifier (Bloom et al., 2017; doi: 10.3334/ORNLDAAC/1502) and will become publicly accessible upon publication. The data availability section has been expanded to include information on the ancillary datasets used in this study.

**Additional changes**

- We identified a minor bug in our code related to the timing MsTMIP heterotrophic respiration outputs; we have updated the results throughout the manuscript accordingly. We note that the outcome had a mininmal impact on the results presented in our manuscript. However, for the sake of clarity, the WetCHARTs code – now included in the supplementary material of the revised manuscript – also includes the subroutines used to read MsTMIP heterotrophic respiration outputs.

– We now report the correct  $\mathbf{CH}_4$  emission units in Figure 3 (Tg/month, instead of Tg/yr).

- In the revised manuscript, Figure 1e now correctly shows the FE 5th - 95th percentile values (in the discussion manuscript, the figure was inadvertently showing FE standard deviation values). A global wetland methane emissions and uncertainty dataset for atmospheric chemical transport models (WetCHARTs version 1.0).

A. Anthony Bloom1, Kevin Bowman1, Meemong Lee1, Alexander J. Turner2, Ronny Schroeder3, John R. Worden1, Richard Weidner1, Kyle C. McDonald1,3, Daniel J. Jacob2.

[revised manuscript text omitted]

2.1 Wetland CH4 emissions & uncertainty

We derive wetland CH4 emissions **F** (mg CH4  $m^{-2} dav^{-1}$ ) at time t and location x as:

10

5

$$F(t,x) = s A(t,x) R(t,x) q_{10}^{\frac{T(t,x)}{10}}$$
(1)

where A(t,x) is the wetland extent fraction, R(t,x) is the C heterotrophic respiration per unit area at time t,  $q_{10}$  T(t,x)/10 is the temperature dependence of the ratio of C respired as CH4 (where  $q_{10}$  is the relative CH4:C respiration for a 10°C increase and T(t,x) is the surface skin temperature) and s is a global scale 15 factor. This empirical parameterization provides first order constraints on the role of carbon, water and temperature variability on the global spatial and temporal variability of wetland CH4 emissions. Variants of the equation 1 parameterization have been used within a range of wetland CH4 emission models (e.g., Hodson et al., 2011, Pickett-Heaps et al., 2011, Bloom et al., 2012; Melton et al., 2013 amongst others).

20

In our approach, wetland CH4 emissions statistics within and across  $0.5^{\circ} \times 0.5^{\circ}$  gridcells are derived based on an ensemble of wetland CH4 emission simulations: the 324-member FE is based on 3 CH4:C temperature dependencies, 9 heterotrophic respiration configurations, 4 wetland extent scenarios and 3 global scale factor configurations  $(3 \times 9 \times 4 \times 3 = 324)$ ; the 18-member EE ensemble is a subset of FE, based on data availability during 2001-2015 (see Table 1 for details).

25

The heterotrophic respiration configurations are derived from 8 terrestrial biosphere models used in the Multi-scale Synthesis and Terrestrial Model Intercomparison Project (MsTMIP BG1 simulations, see Huntzinger et al. 2013 and Wei et al. 2014 for model and experiment details) and the global CARbon DAta-MOdel fraMework (CARDAMOM) terrestrial carbon analysis (Bloom et al., 2016). V1.0 outputs from the MsTMIP are available for the period 1900-2010 (Huntzinger et al., 2016); the CARDAMOM analysis was extended to span 2001-2015 based on the Bloom et al. (2016) methodology (see Appendix A for details). Since MsTMIP and CARDAMOM respiration estimates vary intrinsically

5 as a function of temperature,  $q_{10}$  only accounts for the temperature dependence of the fraction of C respired as CH4. We prescribe three CH4:C temperature dependencies (Table 1) which are broadly equivalent to a ±50% range on the CH4:CO2 temperature dependence reported by Yvon-Durocher et al. (2014).

Here we use two spatial (i = 1,2) and two temporal (j = 1,2) wetland extent parameterizations 10 approaches to represent the uncertainty associated with the role of hydrology on wetland CH4 emissions. Each temporal and spatial wetland extent parameterization,  $A_{i,j}(t,x)$  is derived as:

$$A_{i,j}(t,x) = w_i(x)h_{i,j}(t,x),$$
(2)

15 where  $w_i(x)$  represents the wetland extent fraction, and  $h_{i,i}(t,x)$  represents the temporal variability relative to  $w_i(x)$ .  $w_l(x)$  is the sum of all GLOBCOVER wetland and freshwater land cover types (all flooded, water-logged, and inland water body land-cover types; Bontemps et al., 2011);  $w_2(x)$  is the Global Wetland and Lakes Database (GLWD) maximum recorded wetland and freshwater body extent map by Lehner & Doll (2004).

For  $h_{*,i}(\underline{t,x})$ , we use (a) the Surface WAter Microwave Product Series (SWAMPS) multisatellite surface water product (Schroeder et al., 2015;  $\underline{j}=1$ ), and (b) monthly ERA-interim precipitation  $(\underline{j}=2)$ : for i = 1 (i = 2),  $h_{i,ij}(\underline{t,x})$  is normalized such that mean (maximum)  $h_{i,j}(\underline{t,x})$  is equal to 1. In order avoid physically unrealistic outcomes, we derive  $A_{\underline{l,j}}(\underline{t,x})$  as  $\min\{w_{\underline{l}}(x), \underline{h}_{\underline{l,j}}(\underline{
[revised manuscript text omitted]

- wetland CH4 emission latitudes (10°S 80°N; Figure 2), all mean CH4 model estimates are within the mean standard deviation of observed CH4, except for GC at >60°N and all models at 80°N.

The median site-level correlation (Pearson's r) between observed and model de-trended CH4 concentrations (Figure 9) is highest for BL (0.75), followed by EE (0.74), FE (0.73) and GC (0.72). The median RMSE between observed and model de-trended CH4 concentrations for FE (11.78 ppb) and EE (11.89ppb) are lower than BL (12.42 ppb) and GC (median = 13.27 ppb). FE and EE improvements (relative to GC and BL Pearson's r and RMSE) are primarily in northern hemisphere high-latitudes latitudes (>50°N; Figure 9). In southern hemisphere extra-tropical latitudes (<23°S) FE and EE exhibit a comparable performance relative GC, while BL outperforms both FE and EE.

25

5

**4. Discussion**

**4.1 Model limitations**

Densely vegetated wetland areas are likely to amount to a large component of the global wetland CH4 sources; high-carbon density (and high temperatures in the case of tropical wetlands) result in high CH4 emissions under inundated conditions. However, satellite-derived observations of surface water area

- 5 (Schroeder et al., 2015) are ill-equipped to observe densely vegetated wetland areas, as the passive microwave sensors become increasingly sensitive to vegetation moisture within high biomass ecosystems (Sippel et al., 1994). For example, FE estimates of Amazon river basin wetland CH4 emissions amount to 16% 29% (5th 95th percentiles) of the global wetland emissions source; high biomass density in this region (Saatchi et al., 2011) may be a significant source of inundation area bias.
- 10 Therefore, while we incorporate prior information on the mean and maximum wetland extent to scale the satellite-derived inundation fraction, we anticipate that errors in seasonal and inter-annual inundation variability are likely to be larger within densely vegetated wetland areas. We are optimistic that current and upcoming missions such as SMAP and BIOMASS (Entekhabi et al., 2010; Le Toan et al., 2011) combined with data integration approaches (Schroeder et al., 2015; Fluet-Chouinard et al.,
- 15 2015) can potentially provide additional constraints required to extend current inundation datasets and to improve current surface inundation detection capabilities.

The MsTMIP model ensemble provides a first-order estimate of the magnitude and variability of C decomposition within each 0.5°×0.5° grid-cell. Here we highlight 4 potentially major sources of error:
(a) differences in aerobic:anaerobic turnover rates of major (labile and recalcitrant) C pools (b) systematic differences in wetland and non-inundated area carbon uptake within each 0.5°×0.5° grid-cell, (c) systematic differences in dead organic matter C stocks and accumulation between wetland and non-inundated areas, and (d) lateral flows of C into (or out of) wetland areas. Top-down estimates of seasonal and inter-annual terrestrial CO2 fluxes (e.g. Liu et al., 2014) could be used to independently assess the validity of heterotrophic respiration from the MsTMIP models and CARDAMOM. In turn, top-down CH4 and CO2 flux retrievals, and range of in-situ and regional-scale CH4 flux estimates (Schriel-Uijl et al., 2011; Chang et al., 2014; Budishchev et al., 2014; amongst others) can be combined to assess whether our empirical parameterization is able to capture regional, seasonal and inter-annual

wetland CH4 emission variability and their link to the broader terrestrial carbon cycle. Finally, in succession to eddy covariance tower site analyses of CO2 respiration dependence on temperature (Mahecha et al., 2010), we anticipate that CH4 eddy covariance measurements will provide critical sitelevel constraints on the temperature dependence of wetland CH4 
[revised manuscript text omitted]